# Non-Markovian Reward Modelling from Trajectory Labels via Interpretable Multiple Instance Learning

**Joseph Early** [*†§]     **Tom Bewley** [*‡§]     **Christine Evers** [†]     **Sarvapali Ramchurn** [†§]

## Abstract

We generalise the problem of reward modelling (RM) for reinforcement learning (RL) to handle non-Markovian rewards. Existing work assumes that human evaluators observe each step in a trajectory independently when providing feedback on agent behaviour. In this work, we remove this assumption, extending RM to capture temporal dependencies in human assessment of trajectories. We show how RM can be approached as a multiple instance learning (MIL) problem, where trajectories are treated as bags with return labels, and steps within the trajectories are instances with unseen reward labels. We go on to develop new MIL models that are able to capture the time dependencies in labelled trajectories. We demonstrate on a range of RL tasks that our novel MIL models can reconstruct reward functions to a high level of accuracy, and can be used to train high-performing agent policies.

## 1 Introduction

There is growing consensus around the view that aligned and beneficial AI requires a reframing of objectives as being contingent, uncertain, and learnable via interaction with humans [35]. In reinforcement learning (RL), this proposal has found one formalisation in reward modelling (RM): the inference of agent objectives from human preference information such as demonstrations, pairwise choices, approval labels, and corrections [29]. Prior work in RM typically assumes that a human evaluates the *return* (quality) of a sequential trajectory of agent behaviour by summing equal and independent *reward* assessments of instantaneous states and actions, with the aim of RM being to reconstruct the underlying reward function. However, in reality the human's experience of a trajectory is likely to be temporally extended (e.g., via a video clip [12] or real-time observation), which opens the door to dependencies between earlier events and the assessment of later ones. The independence assumption may be both psychologically unrealistic given human memory limitations [26], and technically naïve given the difficulty of building complete instantaneous state representations [25]. We thus seek to generalise RM to allow for temporal dependencies in human evaluation, by postulating *hidden state* information that accumulates over a trajectory. Reconstruction of the human's preferences now requires the modelling of hidden state dynamics alongside the reward function itself.

In tackling this generalised problem, we identify a structural isomorphism between RM (specifically from trajectory return labels) and the established field of multiple instance learning (MIL) [10]. Trajectories are recast as *bags* and constituent state-action pairs as *instances*, which collectively contribute to labels provided at the bag level by interacting in potentially complex ways. This mapping inspires a range of novel MIL model architectures that use long short-term memory (LSTM) modules [19] to recover the hidden state dynamics, and learn instance-level reward predictions from return-labelled trajectories of arbitrary length. In experiments with synthetic oracle labels, we show that our MIL RM models can accurately reconstruct ground truth hidden states and reward functions for non-Markovian tasks, and can be straightforwardly integrated into RL agent training to achieve performance matching, or even exceeding, that of agents with direct access to true hidden states and rewards. We then apply interpretability analysis to understand what the models have learnt.

---

[*]Equal contribution

[†]University of Southampton, United Kingdom; {J.A.Early,C.Evers,sdr1}@soton.ac.uk

[‡]University of Bristol, United Kingdom; tom.bewley@bristol.ac.uk

[§]The Alan Turing Institute, United Kingdom

36th Conference on Neural Information Processing Systems (NeurIPS 2022).

Our contributions are as follows:

1. We generalise RM to handle *non-Markovian* rewards that depend on hidden features of the environment or the psychology of the human evaluator in addition to visible states/actions.

2. We identify a structural connection between RM and MIL, creating the opportunity to transfer concepts and methods between the two fields.

3. We propose novel LSTM-based MIL models for this generalised RM problem, and develop interpretability techniques for understanding and verifying the learnt reward functions.

4. We compare our proposed models to existing MIL baselines on five non-Markovian tasks, evaluating return prediction, reward prediction, robustness to label noise, and interpretability.

5. We demonstrate that the hidden state and reward predictions of our MIL RM models can be used by RL agents to solve non-Markovian tasks.

The remainder of this work is as follows. Section 2 discusses related work in RM and MIL, Section 3 gives a formal problem definition and describes our MIL-inspired methodology, and Section 4 presents our experiments and results. We discuss key findings in Section 5, and Section 6 concludes. All of our source code is available in a public repository.[1]

## 2 Background and Related Work

**Reward Modelling**   RM [29] aims to infer a reward function from revealed human preference information such as demonstrations [32], pairwise choices [12], corrections [4], good/bad/neutral labels [33], or combinations thereof [24]. Most prior work assumes a human evaluates a trajectory by summing independent rewards for each state-action pair, but in practice their experience is likely to be temporally extended (e.g., via a video clip), creating the opportunity for dependencies to emerge between earlier events and the assessment of later ones. As noted by Chan et al. [11] and Bewley and Lecue [8], such dependencies may arise from cognitive biases such as anchoring, prospect bias, and the peak-end rule [26], but they could equally reflect rational drivers of human preferences not captured by the state representation. Some efforts have been made to model temporal dependencies, such as a discrete psychological mode which evolves over consecutive queries about hypothetical trajectories [6], or a monotonic bias towards more recently-viewed timesteps due to human memory limitations [28]. Elsewhere, Shah et al. [36] use human demonstrations and binary approval labels to learn temporally extended task specifications in logical form. In comparison to these restricted examples, our work provides a more general approach to capturing temporal dependencies in RM.

**Non-Markovian Rewards**   In the canonical RL problem setup of a Markov decision process (MDP), rewards depend only on the most recent state-action pair. In a non-Markovian reward decision process (NMRDP) [2], rewards depend on the full preceding trajectory [2]. NMRDPs can be *expanded* into MDPs (and thus solved by RL) by augmenting the state with a hidden state that captures all reward-relevant historical information, but this is typically not known *a priori*. Data-driven approaches to learning NMRDP expansions [21] often make use of domain-specific propositions and temporal logic operators [3, 39, 41]. Outside of the RM context, recurrent architectures such as LSTMs have been used in NMRDPs to reduce reliance on pre-specified propositions [23]. They also have a long history of use in partially observable MDPs, where dynamics are also non-Markovian [5, 17, 46].

**Multiple Instance Learning**   In MIL [10], datasets are structured as collections of bags $X_i \in \mathbf{X}$, each of which is comprised of instances $\{x_1^i, \ldots, x_k^i\}$ and has an associated bag-level label $Y_i$ and instance-level labels $\{y_1^i, \ldots, y_k^i\}$. The aim is to construct a model that learns solely from bag labels; instance labels are not available during training, but may be used later to evaluate instance-level predictions. The simplest MIL approaches assume that instances are independent and that the bag is unordered, but models exist for capturing various types of instance dependencies [22, 42, 45]. LSTMs have emerged as a natural architecture for modelling temporal dependencies among ordered bags, where they can be utilised to aggregate instance information into an overall bag representation. They have previously been applied to standard MIL benchmarks [44], as well as specific problems such as Chinese painting image classification [30]. As we discuss in Section 3.2, these existing models are somewhat unsuitable for use in RM, leading us to propose our own novel model architectures.

---

[1] https://github.com/JAEarly/MIL-for-Non-Markovian-Reward-Modelling

# 3 Methodology

In this section, we present the core methodology of our work. We formally define the new paradigm of non-Markovian RM (Section 3.1), before drawing on the MIL literature to propose models that can be used to solve this generalised problem (Section 3.2). We then go on to discuss how we can use our learnt RM models for training RL agents on non-Markovian tasks (Section 3.3).

## 3.1 Formal Definition of Non-Markovian RM

Consider an agent interacting with an environment with Markovian dynamics. At discrete time $t$, the current environment state $s_t \in \mathcal{S}$ and agent action $a_t \in \mathcal{A}$ condition the next environment state $s_{t+1}$ according to the dynamics function $D : \mathcal{S} \times \mathcal{A} \to \Delta\mathcal{S}$. A trajectory $\xi \in \Xi$ is a sequence of state-action pairs, $\xi = ((s_0, a_0), ..., (s_{T-1}, a_{T-1}))$, and a human's preferences about agent behaviour respect a real-valued return function $G : \Xi \to \mathbb{R}$. In traditional (Markovian) RM, return is assumed to decompose into a sum of independent rewards over state-action pairs, $G(\xi) = \sum_{t=0}^{T-1} R(s_t, a_t)$, and the aim is to reconstruct $R' \approx R$ from possibly-noisy sources of preference information. In our generalised non-Markovian model, we consider the human to observe a trajectory sequentially and allow for the possibility of hidden state information that accumulates over time and parameterises $R$:

$$G(\xi) = \sum_{t=0}^{T-1} R(s_t, a_t, h_{t+1}) \quad \text{where} \quad h_{t+1} = \delta(h_t, s_t, a_t), \tag{1}$$

$\delta$ is a hidden state dynamics function, and $h_0$ is a fixed value for the initial hidden state. Reconstruction of the human's preferences now requires the estimation of $\delta' \approx \delta$ and $h'_0 \approx h_0$ alongside $R' \approx R$. We visualise the difference between Markovian and non-Markovian RM in Figure 1.

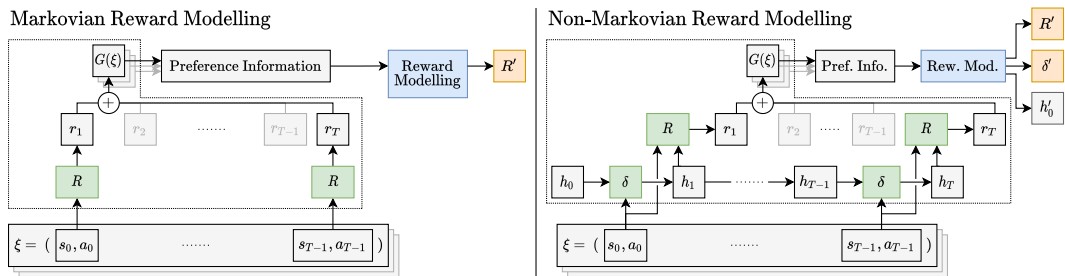

Figure 1: In Markovian RM, the human is assumed to sum $(+)$ over independent and equal reward assessments for the state-action pairs in a trajectory. In non-Markovian RM, per-timestep rewards additionally depend on hidden state information $h$ that accumulates over time.

The hidden state $h$ may be interpreted as (1) an external feature of the environment that is detectable by the human but excluded from the state, or (2) a psychological feature of the person themselves, through which their response to each new observation is influenced by what they have seen already. The latter framing is more interesting for our purposes, and connects to the psychological literature on human judgement, memory, and biases [26]. In practice, hidden state information may encode the human's preferences about the order in which a sequence of behaviours should be performed, the effect of historic observations on their subjective mood (and in turn on their reward evaluations), or cognitive biases which corrupt the way they aggregate instantaneous rewards into trajectory-level feedback. All of these complications are liable to arise in practical RM applications, but cannot be handled when the Markovian reward assumption is made. Appendix B elaborates on this discussion, presenting motivating use cases and limitations of non-Markovian RM.

In this work, we focus on one of the simplest and most explicit forms of preference information: direct labelling of returns $G(\xi_i)$ for a dataset of $N$ trajectories $\{\xi_i\}_{i=1}^N$. We aim to solve the reconstruction problem by minimising the squared error in predicted returns:

$$\underset{R', \delta', h'_0}{\mathrm{argmin}} \sum_{i=1}^N \left( G(\xi_i) - \sum_{t=0}^{T_i-1} R'(s_{i,t}, a_{i,t}, h'_{i,t+1}) \right)^2 \quad \text{where} \quad \begin{matrix} h'_{i,0} = h'_0 \\ h'_{i,t+1} = \delta'(h'_{i,t}, s_{i,t}, a_{i,t}) \end{matrix} \quad \forall i \in \{1..N\}. \tag{2}$$

We observe that Equation 2 perfectly matches the definition of a MIL problem. Each trajectory $\xi_i$ can be considered as an ordered bag of instances $((s_{i,0}, a_{i,0}), ..., (s_{i,T_i-1}, a_{i,T_i-1}))$ with unobserved instance labels $R(s_{i,t}, a_{i,t}, h_{i,t+1})$, an observed bag label $G(\xi_i) = \sum_{t=0}^{T-1} R(s_{i,t}, a_{i,t}, h_{i,t+1})$, and temporal instance interactions via the changing hidden state $h_{i,t}$. This correspondence motivates us to review the space of existing MIL models (specifically those that model temporal dependencies among instances) to provide a starting point for developing our non-Markovian RM approach.

## 3.2 MIL RM Architectures

The MIL literature contains a variety of architectures for handling temporal instance dependencies, including graph neural networks (GNNs) [42] and transformers [37]. While effective for many problems, such architectures are an unnatural fit to non-Markovian RM as they contain no direct analogue of a hidden state $h'$ carried forward in time, instead handling dependencies via some variant of message-passing between instances. LSTM-based MIL architectures [30, 44] provide a more promising starting point since they explicitly represent both $h'$ (implemented as a continuous-valued vector) and its temporal dynamics function $\delta'$ (a particular arrangement of gating functions).

Starting from an existing LSTM-based MIL architecture, we propose two successive extensions as well as a naïve baseline that cannot handle temporal dependencies. All four architectures include a feature extractor (FE) for mapping state-action pairs into feature vectors and a head network (HN) that outputs predictions. These architectures are depicted in Figure 2. Note we use the same nomenclature as [10] and [45]: *embedding space* approaches produce an overall bag representation that is used for prediction, while *instance space* approaches produce predictions for each instance in the bag and then aggregate those predictions to a final bag prediction.

**Base Case: Embedding Space LSTM** This architecture, proposed by Wang et al. [44], processes all instances in a bag sequentially and uses the final LSTM hidden state as a bag embedding. This is fed into the HN, which predicts the bag label $g'$ (return in the RM context). Although this model can account for temporal dependencies, it does not inherently produce instance predictions (rewards), which require some post hoc analysis to recover. While methods exist for computing instance importance values as a form of interpretability [13], these are not guaranteed to sum to the bag label as stipulated by the reward-return formulation. We propose a new method: at time $t$, the predicted reward $r'_t$ is calculated by feeding the LSTM hidden state at times $t-1$ and $t$ into the HN to obtain two *partial* bag labels/returns $g'_{t-1}$ and $g'_t$, and computing the difference of the two, i.e., $r'_t = g'_t - g'_{t-1}$. We define $g'_0 = 0$. This post hoc computation is shown in purple in Figure 2.

**Extension 1: Instance Space LSTM** The post hoc computation of reward proposed above is rather inelegant and often yields poor predictions (see Section 4 and Appendix D), likely because rewards are never computed or back-propagated through during learning. This leads us to propose an improved architecture, which is structurally similar but differs in how network outputs are mapped onto RM concepts. The change places reward predictions on the back-propagation path. Given the LSTM hidden state at time $t$, the output of the HN is taken to be the instantaneous reward $r'_t$ rather than the partial return. Rewards are computed sequentially for all timesteps in a trajectory and summed to give the return prediction $g'$. We thereby obtain a model that both handles temporal dependencies and produces explicitly-learnt reward predictions.

**Extension 2: Concatenated Skip Connection (CSC) Instance Space LSTM** In both of the preceding architectures, the LSTM hidden state $h'_t$ is the sole input to the HN. This requires $h'_t$ to represent all reward-relevant information from both the true hidden state $h_t$ and the latest state-action pair $s_{t-1}, a_{t-1}$ to achieve good performance. To lighten the load on the LSTM, we further extend the Instance Space LSTM model with a skip connection [18, 20] which concatenates the FE output onto the hidden state before feeding it to the HN. In principle, this should allow the hidden state to solely focus on representing temporal dependencies. As well as improving RM performance compared to an equivalent model without skip connections, we find in Section 5.1 that this modification tends to yield more interpretable and disentangled hidden state representations.

**Markovian Baseline: Instance Space Neural Network (NN)** To quantify the cost of ignoring temporal dependencies, we also run experiments with a baseline architecture that feeds only the FE output for each state-action pair into the HN, yielding fully-independent reward predictions which are summed to give the return prediction. This independent predict-and-sum architecture has precedence in both MIL, where it is referred to as mi-Net by Wang et al. [45], and in RM, where it embodies the de facto standard Markovian reward assumption [12].

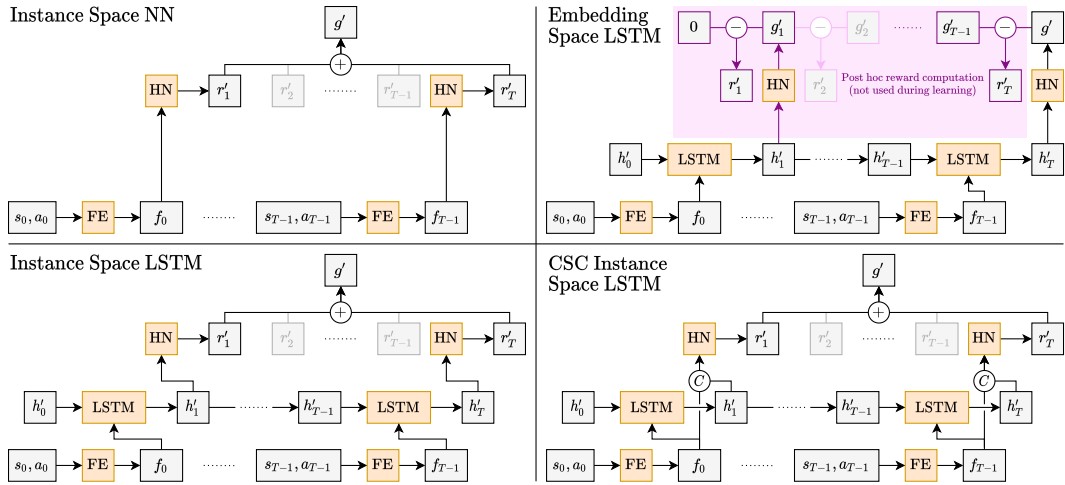

Figure 2: MIL architectures used in this work. FE = feature extractor; HN = head network; $(+)$ = scalar summation; $(-)$ = scalar subtraction; $(C)$ = vector concatenation.

### 3.3 Training Agents with Non-Markovian RM Models

In this work, as in RM more widely, we are not solely interested in learning reward functions to represent human preferences, but also in the downstream application of rewards to train agents' action-selection policies. After optimising our LSTM-based models on offline trajectory datasets, we deploy them at the interface between conventional RL agents and their environments. Going beyond prior work, where a learnt model is used to either generate a reward signal for an agent to maximise [12] or augment its observed state representation with hidden state information [21], our models serve a dual role, providing *both* rewards and state augmentations. Figure 3 describes this setup in detail.

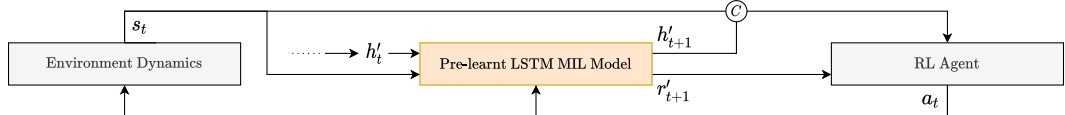

Figure 3: During RL agent training, our LSTM MIL models sit at the centre of the agent-environment loop by which states $s_t$ and actions $a_t$ are exchanged. We focus on episodic tasks, where the environment state periodically resets. The LSTM hidden state is simultaneously reset to $h'_0$ at the start of an episode, then is iteratively updated over time $t$ given the state-action pairs $s_t, a_t$. At time $t$ the environment state $s_t$ is augmented with the post-update hidden state $h'_{t+1}$ by concatenation, and this augmented state is observed by the agent. $s_t$, $a_t$ and $h'_{t+1}$ are used to compute a reward $r'_{t+1}$ following the relevant steps from Figure 2, and the reward is also sent to the agent. In the language of NMRDPs, the hidden state augmentation *expands* the agent's learning problem into an MDP by providing the additional information required to make the rewards Markovian. Note that unlike during learning of the MIL models, return predictions are never required.

## 4 Experiments and Results

After initially validating our models on several toy datasets (see Appendix D), we focus the bulk of our evaluation on five RL tasks. As running experiments with people is costly, we use the standard RM approach of generating synthetic preference data (here trajectory return labels) using ground truth *oracle* reward functions [12] (for a discussion comparing the use of oracle and human labels, see Appendix C). Unlike prior work, these oracle reward functions depend on historical information that cannot be recovered from the instantaneous environmental state, thereby emulating the disparity between the information that a human evaluator possesses while viewing a trajectory sequentially, and that contained in the state alone. In this section, we introduce our RL tasks (Section 4.1), evaluate the quality of reward reconstruction (Section 4.2), investigate the use of MIL RM models for agent training (Section 4.3), and evaluate their robustness to label noise (Section 4.4).

## 4.1 RL Task Descriptions

We apply our methods to five non-Markovian RL tasks, the first four of which are within a common 2D navigation environment and are specifically designed to capture different kinds of non-Markovian structure. Each environment has two spawn zones and an episode time limit of $T = 100$; see Figure 4. In each case, the environment state contains the $x, y$ position of the agent only. The tasks involve moving into a *treasure* zone, contingent on some hidden information that cannot be derived from the current $x, y$ position, but is instead a function of the full preceding trajectory. In the first two cases the hidden information varies with time only, but in the other two it depends on the agent's past positions.

**Timer** For times $t \leq 50$ the treasure gives a reward of $-1$ for each timestep that the agent spends inside it, before switching to $+1$ thereafter. Since time is not included in the environment state, recovering the reward function by only observing the agent's current position is impossible.

**Moving** The Timer task only captures a binary change, therefore we generalise it to be continuous. In this case, the treasure zone oscillates left and right at a constant speed. Again, this is not captured in the environment state, but can be recovered if the length of the preceding trajectory is known.

**Key** Before reaching the treasure zone, the agent must first enter a second zone to collect a key; otherwise it receives 0 reward. As the key's status is not captured in the environment state, a temporal dependency exists between the agent's past positions and the reward it obtains from the treasure.

**Charger** We generalise the Key task by replacing the key zone with a charging zone that builds up the amount of reward the agent will receive when it reaches the treasure. The reward now depends not only on whether the agent visits a zone (binary), but how long it spends there (continuous).

For the fifth and most complex task, we adapt **Lunar Lander** from OpenAI Gym [9], adding the condition that the lander should take off again and stably hover after 50 timesteps on the landing pad. This is analogous to the Charger task but with a larger state-action space and longer episodes ($T = 500$). Further details on the tasks and MIL model hyperparameters are given in Appendix E.

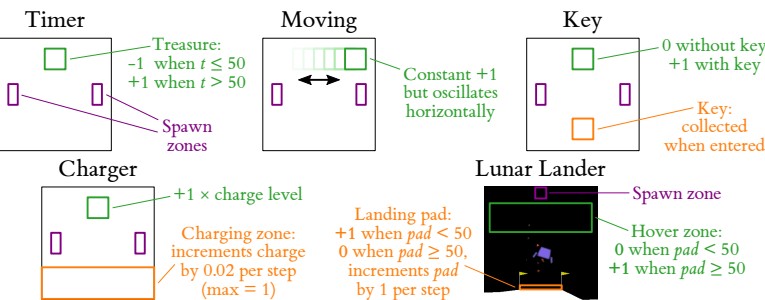

Figure 4: Visualisations of the five non-Markovian RL tasks.

An important design decision for the LSTM-based models is the size of the hidden state, as it affects both performance and interpretability. For all the above tasks, we know *a priori* that it is possible to capture the temporal dependencies in at most two dimensions, so we constrain our models to use 2D hidden states. This allows us to visualise and interpret the hidden representations in Section 5.1.

## 4.2 Reward Modelling Results

Below we discuss the performance of the reward reconstruction for the different MIL RM models on our five RL tasks. For each task, we generate initial trajectories to form our MIL RM datasets (see Appendix E). Results from MIL models trained on these trajectories are given in Table 1. We observe that the CSC Instance Space LSTM model is on average the best-performing model for predicting both trajectory returns and timestep rewards. While the Embedding Space LSTM model performs best at predicting return on the Key and Lunar Lander tasks (as is not constrained to the summation of reward predictions as in the other architectures), it struggles on the reward metric (due to the use of a proxy post hoc method). As it is important for these models to achieve strong performance on both return and reward prediction, the Instance Space LSTM and CSC Instance Space LSTM models are better candidates than the Embedding Space LSTM. Also note that the Instance Space NN that serves as our Markovian RM baseline performs very poorly on return prediction, indicating that these tasks indeed cannot be learnt without modelling temporal dependencies.

Table 1: MIL RM return (top) and reward (bottom) results, using ten repeats. The Lunar Lander results are the average test set MSE of the top five models (with scaling; see Appendices E.4 and E.5). For the other tasks, each measurement is the test set MSE averaged over all ten repeats. The standard errors of the mean are given, and the Lunar Lander reward results are scaled by $1 \times 10^{-5}$.

| Model | Timer | Moving | Key | Charger | Lunar Lander |
|---|---|---|---|---|---|
| Instance Space NN | $130.8 \pm 1.530$ | $22.24 \pm 0.441$ | $7.764 \pm 0.232$ | $7.783 \pm 0.214$ | $2.297 \pm 0.058$ |
| Embedding Space LSTM | $3.151 \pm 0.662$ | $13.04 \pm 0.899$ | $\mathbf{0.360 \pm 0.055}$ | $0.689 \pm 0.124$ | $\mathbf{0.416 \pm 0.048}$ |
| Instance Space LSTM | $7.313 \pm 2.627$ | $11.13 \pm 1.169$ | $0.488 \pm 0.062$ | $0.628 \pm 0.126$ | $1.223 \pm 0.431$ |
| CSC Instance Space LSTM | $\mathbf{0.605 \pm 0.166}$ | $\mathbf{5.307 \pm 0.299}$ | $0.391 \pm 0.083$ | $\mathbf{0.125 \pm 0.012}$ | $0.501 \pm 0.035$ |
| Instance Space NN | $0.217 \pm 0.001$ | $0.068 \pm 0.000$ | $0.011 \pm 0.000$ | $0.025 \pm 0.000$ | $7.484 \pm 0.861$ |
| Embedding Space LSTM | $101.8 \pm 60.35$ | $3.033 \pm 0.715$ | $0.010 \pm 0.008$ | $0.037 \pm 0.016$ | $120.2 \pm 24.27$ |
| Instance Space LSTM | $0.263 \pm 0.038$ | $0.069 \pm 0.005$ | $0.002 \pm 0.000$ | $0.005 \pm 0.001$ | $9.336 \pm 3.116$ |
| CSC Instance Space LSTM | $\mathbf{0.073 \pm 0.016}$ | $\mathbf{0.026 \pm 0.002}$ | $\mathbf{0.001 \pm 0.000}$ | $\mathbf{0.001 \pm 0.000}$ | $\mathbf{7.365 \pm 1.032}$ |

### 4.3 RL Training Results

Following the method in Section 3.3, we then train Soft Actor-Critic [15] (Lunar Lander) and Deep Q-Network [31] (all others) RL agents to optimise the rewards learnt by the LSTM-based models. We evaluate agent performance in a post hoc manner by passing its trajectories to the relevant oracle. This evaluation provides an end-to-end measure of both reward reconstruction and policy learning, and is standard in RM [12]. We baseline against agents trained with access to: a) the oracle reward function and the oracle hidden states, and b) just the oracle reward function without hidden states (i.e, using only the environment states that are missing information). In Figure 5, we observe that the CSC Instance Space LSTM model enables the best RL agent performance, coming closest to the oracle. Interestingly, for the Timer and Lunar Lander tasks, the CSC Instance Space LSTM model actually outperforms the use of the oracle, suggesting that the learnt hidden states are easier to exploit for policy learning than the raw oracle state (we investigate what these models have learnt in Section 5.1). Note the poor performance of agents trained without hidden state information, which aligns with expectations. For further details on agent training, see Appendix F.

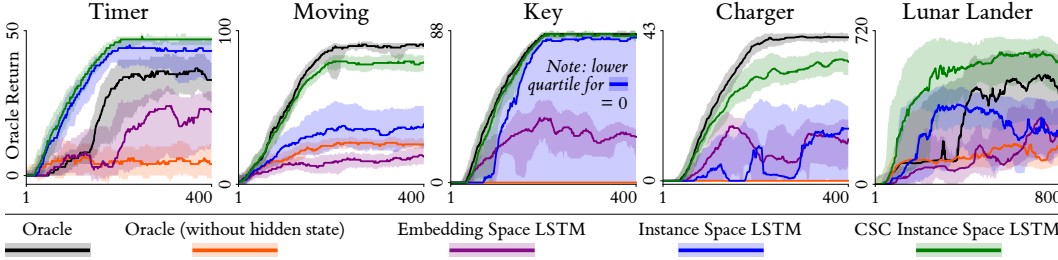

Figure 5: RL performance for different training configurations on our five RL tasks. The results given are the medians and interquartile ranges. For the oracle results, we trained ten repeats, and for the MIL-LSTM results, we performed one RL training run for each MIL-LSTM model repeat.

For Lunar Lander, we perform a deeper analysis of RL training performance, by decomposing the oracle return curves from Figure 5 into the four reward components $R_{pad}$, $R_{no\_contact}$, $R_{hover}$ and $R_{shaping}$ (see Appendix E.1 for definitions). The decomposed curves, shown in Figure 6, allow us to diagnose the origins of the performance disparity between runs using different LSTM model architectures. There is relatively little separation in performance on the shaping reward $R_{shaping}$ and pad contact reward $R_{pad}$ (for the latter, all runs end up reliably achieving the maximum possible reward of 49, although those using Embedding Space LSTM models require significantly more training time). This suggests that all models have been able to recover these components with reasonable fidelity. However, there are marked differences in performance on $R_{no\_contact}$ and $R_{hover}$ (the components relating to the second task stage of taking off and moving to the hover zone). For $R_{hover}$, runs using the CSC Instance Space LSTM peak at a return of around 200 from this component, while those using the other two models almost never achieve non-zero return, i.e., only the RL agents trained using the CSC Instance Space LSTM RM models reliably learn to hover. This indicates that the models have learnt very different representations of reward and hidden state dynamics, which are effective for policy learning in the case of the CSC model, and highly ineffective for the others.

Observe that runs using CSC Instance Space LSTM models outperform those with direct access to the ground truth oracle on all components, and most markedly on $R_{\text{hover}}$. This counterintuitive finding suggests that this model reliably learns hidden state representations that are easier for RL agents to leverage for policy learning than the ground truth ones, and potentially that certain errors in the reward prediction may actually be beneficial for the purpose of helping agents to complete the underlying task (especially the hovering stage). In typical RL parlance: the model's reward function appears to be better *shaped* than the ground truth. The potential origins of this better-than-oracle phenomenon are investigated in Figure A5 (Appendix G).

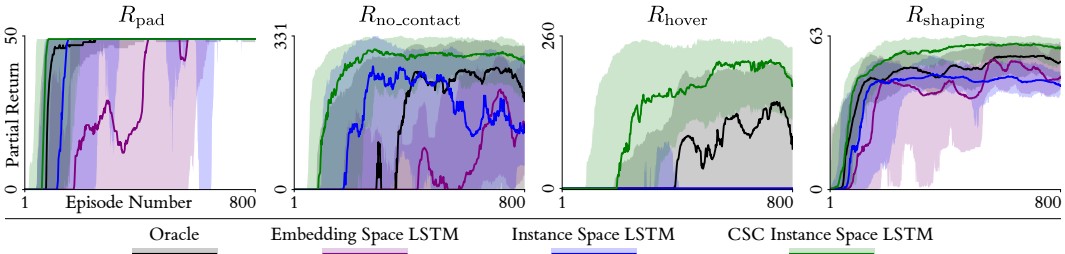

Figure 6: Decomposed oracle return curves for Lunar Lander.

## 4.4 Robustness to Mislabelling

In this work, the return labels are provided by oracles rather than real people. When using human evaluators, there is likely to be uncertainty in the labels, and it is important to evaluate model robustness against such noise [28]. We implement noise through label swapping [34]; this ensures the marginal label distribution remains the same and does not include out-of-distribution returns. In Figure 7, we show how both return and reward prediction decay with noise levels increasing from 0 (no labels swapped) to 0.5 (half swapped). The rate, smoothness, and consistency (across three repeats) of this decay varies between tasks, with decays in return prediction generally being smoother. We observe that the CSC Instance Space LSTM model remains the strongest predictor of both return and reward in the majority of cases, indicating general robustness and providing evidence that the model should still be effective with imperfect human labels. On all metrics aside from Timer reward loss (where the mix of negative and positive rewards makes the effect of noise especially unpredictable), a noise level of at least 0.3 is required for the CSC Instance Space LSTM model to perform as badly as the Instance Space NN baseline does with no noise at all.

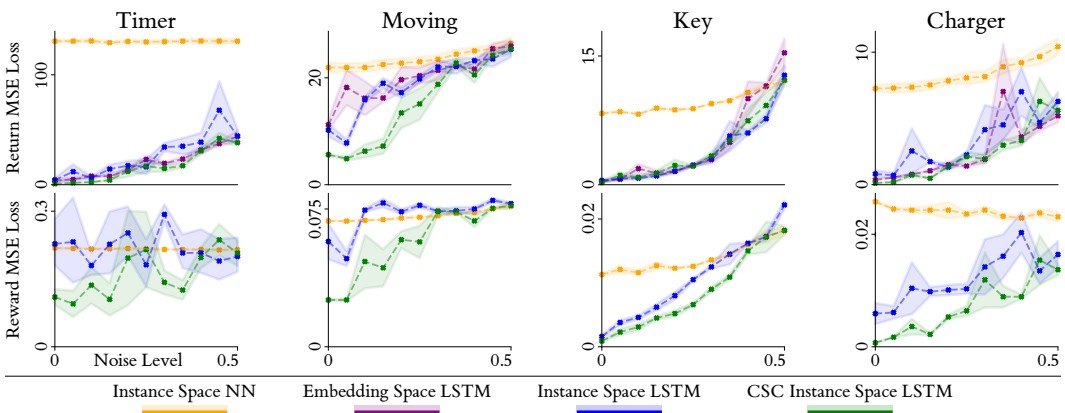

Figure 7: Performance of MIL RM models subject to label noise. We omit the Embedding Space LSTM reward losses as they are very high, and the Lunar Lander task due to long training times.

## 5 Discussion

In this section, we seek to interpret our MIL RM models, analysing the distribution of learnt hidden states (Section 5.1) as well as their temporal dynamics over the course of a trajectory (Section 5.2). Finally, in Section 5.3 we discuss the limitations of this work and potential areas for future work.

## 5.1 Hidden State Analysis

The primary purpose of RM is to perform accurate reward reconstruction to facilitate agent training, but there is a secondary opportunity to improve understanding of human preferences through interpretability analysis of the learnt models. We can directly visualise the 2D LSTM hidden states of our oracle experiments, which enables a qualitative comparison of the various model architectures (see Figure 8). Visualising the hidden states with respect to the temporal dependencies indicates that the CSC Instance Space LSTM model has learnt insightful hidden state representations. Breaking down the CSC Instance Space LSTM model hidden embeddings: for the Timer task, time is represented along a curve, with a sparser representation around $t = 50$ (the crossover point when the treasure becomes positive). For the Moving task, time is similarly captured along with an additional notion of the change in treasure direction from right to left. For the Key task, the binary state of no key vs key is separated, with additional partitioning based on $x$ position, denoting the two different start points of the agent. In the Lunar Lander task, the model has learnt a strong separation between states either side of the crossover point when the time on the pad is equal to $50$, with high sparsity around the crossover point. In comparison, the Embedding and Instance Space LSTM models have not learnt as sparse a representation. We discuss the Charger task in Section 5.2.

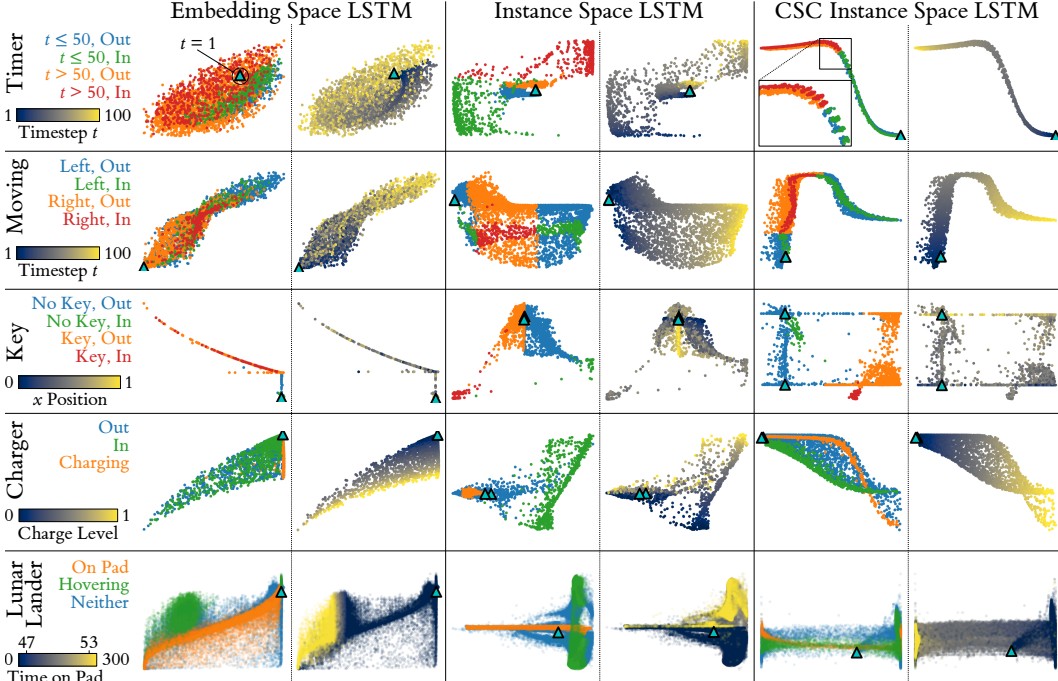

Figure 8: Learnt hidden state embeddings for our MIL RM models. For each model and task, we categorise the hidden state embeddings depending on the true environment state (first column for each model). In and Out environments states indicate whether the agent is in the treasure zone or not, and for the Moving task, Left and Right indicate the direction in which the goal is currently moving. We also provide labelling based on temporal information (second column for each model). Furthermore, we include markers to indicate the hidden states for the centres of the agent spawn zones. In each case, we elected to use the best-performing repeat for each model as assessed by the reward reconstruction (see Table 1). Note, for the Key task, as the temporal information is captured in the state categorisation (No Key vs Key), we use the second column to show the relationship between the hidden embeddings and the agent's $x$ position.

## 5.2 Trajectory Probing

We further interpret our models by visualising the learnt reward with respect to the environment state, and by using hand-specified *probe* trajectories to verify that the learnt hidden state transitions mimic the true transitions. We present the above for the CSC Instance Space LSTM model on the Charger task in Figure 9 (Appendix G contains similar figures for all other tasks). The top row shows that the model has correctly learnt the relationships between position, charge, and reward (reward increases

in the treasure zone as charge increases). From the probes, we can see how the charge level can be recovered from the hidden states. We also note that the inflection point between under-charging and over-charging is captured, i.e., this is where the optimal charge level lies, subject to some noise based on where the agent starts in the spawn zones. Furthermore, with the Challenging probe, we observe that the learnt hidden states align with the agent moving in and out of the charging zone.

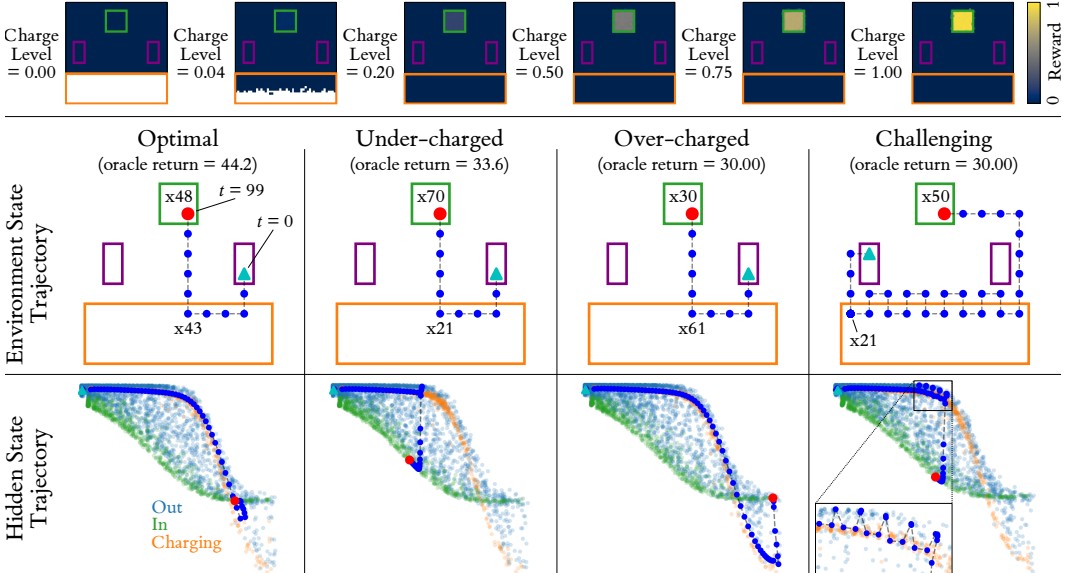

Figure 9: Interpretability for the CSC Instance Space LSTM model on the Charger task. **Top**: the learnt relationship between agent position, charge level, and reward. **Middle/bottom**: Four probe trajectories demonstrating hidden state transitions as the trajectory progresses. *Optimal*: best possible return (charge to sufficient level then maximise time in treasure). *Over-charged*: continuing to charge after maximum charge of 1 is reached. "x$n$" labels indicate the agent remains in a position for $n$ timesteps. As in Figure 8, we use the best-performing model according to reward reconstruction.

## 5.3 Limitations and Future Work

Although we analyse the performance of our methods in the presence of noisy labels in Section 4.4, a major area of future work is to apply our methods to human labelling (for a discussion of this, see Appendix C). Another area of future work involves more complex environments, for example the use of tasks with image observations, similar to the Atari environments in Open AI Gym [9]. Furthermore, we perform RM from either an offline dataset or from only one RL training iteration; an iterative bootstrapping approach with multiple RL + RM training iterations could lead to improved RL results. There are also limitations with our MIL RM approach for the Lunar Lander task; see Appendix E.5 for details and suggestions for future work. More generally, we hope that our identification of the link between RM and MIL may inspire a bidirectional transfer of tools and techniques.

## 6 Conclusion

We posed the problem of non-Markovian RM, which removes an unrealistic assumption about how humans evaluate temporally extended agent behaviours. After identifying an isomorphism between RM and MIL, we proposed and evaluated novel MIL-inspired models that allow us to reconstruct non-Markovian reward functions, augment agent training, and interpret their learnt representations.

## Acknowledgements

This work was funded by the AXA Research Fund, the UKRI Trustworthy Autonomous Systems Hub (EP/V00784X/1), and an EPSRC/Thales industrial CASE award. We would also like to thank the Universities of Southampton and Bristol, as well as the Alan Turing Institute, for their support. The authors acknowledge the use of the IRIDIS (Southampton) and BlueCrystal (Bristol) high-performance computing facilities and associated support services in the completion of this work.

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
