# A   Implementation and Resource Details

This work was implemented in Python 3.8 / 3.10 and the machine learning functionality used PyTorch. All required libraries for our work are given in a `requirements.txt` file. Our code is publicly accessible at https://github.com/JAEarly/MIL-for-Non-Markovian-Reward-Modelling. The majority of MIL model training was carried out on a remote GPU service using a Volta V100 Enterprise Compute GPU with 16GB of VRAM, which utilised CUDA v11.0 to enable GPU support (IRIDIS 5, University of Southampton). For the Lunar Lander task, training each MIL model took a maximum of eight hours. For the other tasks, this was a maximum of two hours. Trained models are included alongside the code. Fixed seeds were used to ensure consistency of dataset splits between training and testing; these are included in the scripts that are used to run the experiments. All our datasets were generated from code; both the scripts to generate the data and also the derived datasets themselves are included alongside our model training code. Dataset generation, as well as all RL agent training, was conducted on a second remote GPU service using a compute node with two Nvidia Pascal P100 cards. Data generation took a maximum of three hours per dataset (BlueCrystal Phase 4, University of Bristol). Agent training for the 2D navigation tasks was computationally light, requiring 8-12 minutes per 400-episode run, although we completed ten repeat runs for each permutation of task and MIL model architecture (one per MIL training repeat). 800-episode RL runs for Lunar Lander took approximately two hours each. Further details on executing the scripts to reproduce our results can be found in the `README.md` file in our code repository.

# B   Use Cases for Non-Markovian Reward Modelling

The non-Markovian reward formulation applies to cases where rewards depend on hidden state information $h_t$ in addition to environment states $s_t$ and actions $a_t$, and this information is a function of previous state-action pairs but *not* vice versa (i.e., there is no causal path from $h_t$ to $s_{t+k}$ for any $k > 1$). This crucial caveat distinguishes the formulation from the more general class of partially observable MDPs and demarcates the set of domains to which it can be applied: those involving a secondary Markovian system that "spectates" on events in the environment without intervening. In the RM context, this secondary system is a black box (making its internal state $h_t$ hidden) and explicit rewards are unavailable, being replaced by a sparser and potentially noisy form of reward-dependent feedback (trajectory return labels in our work). Below we identify three classes of use case which fit this technical specification and provide one concrete example for each:

**Ambiguous Subtasks**   Cases involving an extended task with a sequential structure, where it is hard to formally define the conditions for subtask completion, but RM is feasible because a human "knows it when they see it". Here, the hidden state to be learnt represents the current subtask and any auxiliary information needed to determine its completion status.

  - **Concrete Example:** Using judges' scores to learn a performative display (e.g., gymnastics, aerobatics) chaining several manoeuvres whose start and end conditions are difficult to formalise *a priori*. This could be considered as an extension of the single backflip task studied in the foundational RM work by Christiano et al. [12].

**Dependencies on Subjective Affect**   Cases where a human's reward function is dependent on their affective (emotional) status, which in turn depends on their prior experiences. Assuming this information is not directly available in the observed environment state, it must be inferred from data.

  - **Concrete Example:** Using periodic satisfaction ratings to train a personal assistant robot whose owner's mood, needs and preferences vary from day to day. These variations may influence the preferred driving style of a chauffeur service or choice of evening meal.

**Irrationalities/Cognitive Biases**   Cases where one or more forms of bias colour a human's post hoc rating of an observed trajectory, even if their instantaneously-experienced reward is Markovian. Psychological studies of how humans aggregate immediate rewards into retrospective evaluations of the quality of an experience find that a straight summation assumption is unrealistic, with subjects exhibiting high sensitivity to contrast effects and recency bias (collectively termed the peak-end rule)

[26], and factoring in anticipated future states in addition to those actually observed [1]. Here, the hidden state captures an aggregate representation of the biases at play in a given human's evaluation.[5]

- **Concrete Example:** Using customer "star ratings" to improve a holiday planning agent whose recommendations aim to account for an unknown mix of biases such as the peak-end rule. The agent may use the learnt bias model to prioritise key moments in a holiday when managing the travel schedule and budget, in order to maximise future customers' star ratings.

Finally, we note that it is a matter of taste as to whether hidden state information is framed as situated *inside a human evaluator's mind* or *in the environment but only visible to the human*. It is not technically necessary to decide between these two framings, as the mathematical problem of non-Markovian RM is equivalent. From an agent's perspective, a human evaluator is part of an augmented environment, even if they never intervene directly to influence the state.

## C   Comparing Human and Oracle Labelling

Oracle-based experiments are often used to evaluate RM methods since they enable scalable quantitative validation [14, 16, 33]. However, we identify three concrete differences between our oracle preference labelling method and realistic human labelling: 1) preference form, 2) preference sparsity, and 3) preference noise. Preference form is the different ways of providing labels; in this work we used return values which are highly informative and easy to learn from, but it has long been understood that humans find it easier to give less direct feedback, such as pairwise rankings [12] or good/bad/neural labels [27]. Preference sparsity occurs when the time- and cost-expensiveness of eliciting human labels reduces the proportion of data that can be labelled, and preference noise arises from uncertainties in the human labelling process (as opposed to perfect oracle labelling). We decided to focus on noise in this work as it is an established way of making oracle experiments more realistic [28], and also fits in with our discussion of human uncertainties and cognitive biases. Our experiments in Section 4.4 indicate that our methods degrade gracefully in the presence of noise, which gives us some confidence that they will transfer well to human labels. However, future work should consider preference sparsity and form, the latter of which will involve modifying the data collection pipeline and loss function (e.g., to a contrastive loss in the case of pairwise rankings). Beyond accounting for these three differences whilst still using oracle labels, the next step would be to conduct evaluations using actual human labels.

## D   Preliminary Experiments on Toy Datasets

In this section, we give detail our preliminary experiments that were run on toy problems to initially develop and validate our approach. Below we outline the datasets (Section D.1), models and training hyperparameters (Section D.2), and results (Section D.3) of these experiments.

### D.1   Datasets

We introduce three toy datasets, each abstracted from the RL context, to act as benchmark tests for our models. Each of these datasets uses ordered bags comprised of two-dimensional instances, where each instance has an associated label (called "reward" below, for consistency), and the overall bag label (return) is the sum of the instance labels.

**Toggle Switch**   An instance is the position of a toggle switch $ts$ and a value $v$; if the switch is on ($ts = 1$), then the reward for the instance is $v$; otherwise the reward for the instance is 0. Here, there

---

[5]A fascinating philosophical question arises here. When (following the method of Section 3.3) an RL agent is trained using a non-Markovian RM model that captures a cognitive bias, should the agent learn to maximise rewards *including* the bias (which, for example, might lead it to prioritise its peak and final reward rather than seek uniformly good performance), or *exluding* it (which would revert to uniform prioritisation). This issue of whether intelligent agents should seek to exploit human irrationalities when optimising for their revealed preferences, or appeal to their unbiased "better angels", relates to distinctions between first- and second-order preferences [38] or between experienced and remembered (or decision) utility [7]. We defer this question to those with more relevant expertise but note that regardless of the answer, it is essential to include the biases in the reward model using a method such as the one proposed in this work.

is no hidden information, as it is possible to calculate the reward for an instance from its contents $ts, v$ alone. This serves as a null example to elucidate what happens in a Markovian setting.

**Push Switch** We modify the toggle switch setup so that the instance now represents a push switch ($p = 1$ if this switch is pressed), where pressing the switch flips a binary hidden state $ts$ (the same information as was previously represented by the toggle switch). This hidden state then determines the reward as before ($v$ if $ts = 0$, else 0). As $ts$ must be tracked between successive instances, it is not possible to determine the reward for an instance solely by observing its contents $p, v$, so this setup is non-Markovian.

**Dial** We generalise the hidden state from a binary switch to a continuous-valued dial. Given an instance $m, v$, the dial's current value $d$ is moved up or down by $m$. The reward is then given as $d \cdot v$ The problem remains non-Markovian, but now the hidden state that needs tracking, $d$, is continuous rather than discrete.

## D.2 Models and Hyperparameters

When training the MIL models on the toy dataset, we used the Adam optimiser with a batch size of one (i.e., one bag per batch) to minimise mean squared error (MSE) loss. Training was performed using validation loss early stopping, i.e., if the validation loss did not decrease after a certain number of training epochs (patience value), we terminated the training and selected the model at which the validation loss was lowest. If the patience value was not reached (i.e., the validation loss kept decreasing), we terminated training after a maximum number of epochs had been reached, and again selected the model at which the validation loss was lowest. The hyperparameters for training the models on each dataset (including learning rate (LR) and weight decay (WD)) are given in Table A1. Dropout was not used. These hyperparameters were found through a small amount of trial and error, i.e., no formal hyperparameter tuning was carried out.

Table A1: Toy dataset MIL training hyperparameters.

| Dataset | LR | WD | Patience | Epochs |
|---|---|---|---|---|
| Toggle Switch | $1 \times 10^{-4}$ | $1 \times 10^{-5}$ | 20 | 100 |
| Push Switch | $1 \times 10^{-3}$ | 0 | 30 | 150 |
| Dial | $1 \times 10^{-3}$ | 0 | 30 | 150 |

In Tables A2 to A5 we give the architectures for the MIL models we used in the toy dataset experiments. The models are a combination of fully connected layers (FC) along with different MIL pooling mechanisms. Rectified linear unit (ReLU) activation is applied to the FC hidden layers. We label the layers based on the part of the network they belong to: feature extractor (FE), head network (HN), or pooling (P); see Section 3.2. We also indicate the input and output sizes: $b$ x $n$ indicates an input or output where there is a representation of length $n$ for each of the $b$ instances ($b$ is the size of the input bag).

Table A2: Toy Instance Space NN

| Layer | Type | Input | Output |
|---|---|---|---|
| 1 (FE) | FC + ReLU | $b$ x 2 | $b$ x 2 |
| 2 (HN) | FC | $b$ x 2 | $b$ x 1 |
| 3 (P) | mil-sum | $b$ x 1 | 1 |

Table A3: Toy Embedding Space LSTM

| Layer | Type | Input | Output |
|---|---|---|---|
| 1 (FE) | FC + ReLU | $b$ x 2 | $b$ x 2 |
| 2 (P) | mil-emb-lstm | $b$ x 2 | 2 |
| 3 (HN) | FC | 2 | 1 |

## D.3 Results

For each of the toy datasets, we generate 5000 random bags with between 10 and 20 instances per bag (uniformly distributed). We use an 80/10/10 dataset split for training, validation, and testing, and repeat our experiments with ten different variations of this split (so in total we have ten repeats of each model type for each dataset). We show results for both return and reward reconstruction for the toy datasets in Table A6. From these results, we can make several observations. Firstly, as

| Table A4: Toy Instance Space LSTM | | | | | Table A5: Toy CSC Instance Space LSTM | | | |

| Layer | Type | Input | Output |
|---|---|---|---|
| 1 (FE) | FC + ReLU | $b$ x 2 | $b$ x 2 |
| 2 (P-1) | mil-ins-lstm | $b$ x 2 | $b$ x 2 |
| 3 (HN) | FC | $b$ x 2 | $b$ x 1 |
| 4 (P-2) | mil-sum | $b$ x 1 | 1 |

| Layer | Type | Input | Output |
|---|---|---|---|
| 1 (FE) | FC + ReLU | $b$ x 2 | $b$ x 2 |
| 2 (P-1) | mil-csc-ins-lstm | $b$ x 2 | $b$ x 2 |
| 3 (HN) | FC | $b$ x 2 | $b$ x 1 |
| 4 (P-2) | mil-sum | $b$ x 1 | 1 |

expected, the Instance Space NN architecture only works on the Markovian Toggle Switch dataset, i.e., it fails on the non-Markovian Push Switch and Dial datasets as it is unable to deal with temporal dependencies. We also note that our two proposed architectures (Instance Space LSTM and CSC Instance Space LSTM) outperform the baseline Embedding Space LSTM method on both return and reward, with the CSC Instance Space LSTM model providing the best results overall. Finally, we observe that a better return performance does not always guarantee better reward performance: for the Dial dataset, the Instance Space LSTM makes better return predictions than the CSC Instance Space LSTM model, but worse reward predictions. A similar outcome can be seen for the Embedding Space LSTM and the Instance Space NN on the Push Switch dataset.

Table A6: Toy dataset return (top) and reward (bottom) results. Each measurement is the mean MSE averaged over ten repeats, with the standard errors of the mean also given. Bold entries indicate the best-performing model for each (metric, dataset) pair.

| Model | Toggle Switch | Push Switch | Dial | Overall |
|---|---|---|---|---|
| Instance Space NN | $0.030 \pm 0.029$ | $3.337 \pm 0.054$ | $5.489 \pm 0.157$ | 2.952 |
| Embedding Space LSTM | $0.008 \pm 0.002$ | $0.663 \pm 0.194$ | $0.434 \pm 0.075$ | 0.368 |
| Instance Space LSTM | $0.062 \pm 0.058$ | $0.262 \pm 0.154$ | $\mathbf{0.111 \pm 0.014}$ | 0.145 |
| CSC Instance Space LSTM | $\mathbf{0.000 \pm 0.000}$ | $\mathbf{0.140 \pm 0.065}$ | $0.121 \pm 0.043$ | $\mathbf{0.087}$ |
| Instance Space NN | $0.002 \pm 0.002$ | $0.086 \pm 0.001$ | $0.954 \pm 0.011$ | 0.347 |
| Embedding Space LSTM | $0.003 \pm 0.001$ | $0.206 \pm 0.100$ | $0.244 \pm 0.077$ | 0.151 |
| Instance Space LSTM | $0.004 \pm 0.004$ | $0.021 \pm 0.008$ | $0.026 \pm 0.004$ | 0.017 |
| CSC Instance Space LSTM | $\mathbf{0.000 \pm 0.000}$ | $\mathbf{0.012 \pm 0.004}$ | $\mathbf{0.022 \pm 0.007}$ | $\mathbf{0.011}$ |

# E  RL Task Details, Data Generation, and Model Hyperparameters

In this section, we give more detail on the reward reconstruction experiments for the RL tasks. First, we give more information about the RL tasks (Section E.1), then explain how we generated datasets from the tasks (Section E.2), and give the MIL model architectures and hyperparameters used in the Timer, Moving, Key and Charger tasks (Section E.3), and the Lunar Lander task (Section E.4). Finally, we discuss the limitations of our approach to using MIL RM for the Lunar Lander task (Section E.5).

## E.1  Task Details

The first four tasks are implemented in Python within a common 2D simulator following the OpenAI Gym standard [9]. The agent's position $x, y$ is moved by one of five discrete actions: up, down, left, right and no-op. In the first four cases, the position is moved by $0.1$ in the specified direction. The motion vector is then corrupted by zero-mean Gaussian noise with a standard deviation of $0.02$ in both $x$ and $y$ and clipped into the bounds $[0, 1]^2$. Zones of interest (spawn zones, treasure, key, charger) are specified as rectangles lying within these bounds. At time $t$, the environment state $s_t$ (which is directly observed by the MIL RM models) is the 2D vector of the current position $[x_t, y_t]$; its dynamics are Markovian given the agent's chosen action. The hidden state $h_t$ is the task-specific information that renders the oracle's reward function $R$ Markovian:

- **Timer:** $h_0 = 0$ and $h_{t+1} = \delta(h_t, s_t, a_t) = h_t + 1$; the hidden state simply tracks the current timestep index. Reward is given by[6]

$$R(s_t, a_t, h_{t+1}) = \text{in\_treasure}(s_t) \cdot \begin{cases} -1 & \text{if } h_{t+1} \leq 50, \\ +1 & \text{otherwise,} \end{cases}$$

where $\text{in\_treasure}([x_t, y_t]) = 1$ if $0.4 \leq x_t \leq 0.6$ and $0.7 \leq y_t \leq 0.9$, and 0 otherwise.

- **Moving:** $h_0 = [0.4, -0.02]$, the initial horizontal position (left edge) and velocity of the moving treasure rectangle. Hidden state dynamics encode the left-right oscillation:

$$h_{t+1} = \delta(h_t, s_t, a_t) = \left[ h_t^0 + h_t^1, \begin{cases} h_t^1 & \text{if } 0 < (h_t^0 + h_t^1) < 0.8, \\ -h_t^1 & \text{otherwise} \end{cases} \right].$$

Reward is given by $R([x_t, y_t], a_t, h_{t+1}) = 1$ if $h_{t+1} \leq x_t \leq (h_{t+1} + 0.2)$ and $0.7 \leq y_t \leq 0.9$, and 0 otherwise.

- **Key:** $h_0 = 0$, indicating that the agent initialises without the key. The key collection dynamics are encoded by

$$h_{t+1} = \delta(h_t, [x_t, y_t], a_t) = \begin{cases} 1 & \text{if } 0.4 \leq x_t \leq 0.6 \text{ and } 0.1 \leq y_t \leq 0.3, \\ h_t & \text{otherwise.} \end{cases}$$

Reward is given by $R(s_t, a_t, h_{t+1}) = \text{in\_treasure}(s_t) \cdot h_{t+1}$, where the in\_treasure function is the same as in the Timer task.

- **Charger:** $h_0 = 0$, indicating an initial charge level of zero. The charging dynamics are encoded by

$$h_{t+1} = \delta(h_t, [x_t, y_t], a_t) = \begin{cases} \min(h_t + 0.02, 1) & \text{if } y_t \leq 0.3, \\ h_t & \text{otherwise.} \end{cases}$$

Reward is given identically to the Key task, $R(s_t, a_t, h_{t+1}) = \text{in\_treasure}(s_t) \cdot h_{t+1}$.

The **Lunar Lander** task is a modified version of the LUNARLANDERCONTINUOUS-V2 baseline included as standard in the OpenAI Gym library [9]. We leave the state and action spaces unmodified. The 8D state vector is $[x, y, v^x, v^y, \theta, \dot{\theta}, c^l, c^r]$, where $x, y$ and $v^x, v^y$ are the landing craft's horizontal and vertical positions and velocities, $\theta$ and $\dot{\theta}$ are its angle from vertical and angular velocity, and $c^l, c^r$ are two binary contact detectors indicating whether the left and right landing legs are in contact with the ground. The 2D continuous action $[u^m, u^s]$ is a pair of throttle values for two engines: main $u^m$ and side $u^s$. We also retain the default initialisation conditions (the lander spawns in a narrow zone above the landing pad, with slightly-randomised orientation and velocities), the automatic termination of episodes when $|x|$ exceeds 1 (i.e., when the lander leaves the rendered screen area), and the physics that determine how the lander responds to engine activations. However, we replace the standard reward function with an oracle that rewards the agent for landing on the pad for up to 50 timesteps, and then taking off again to hover within a target zone until an episode time limit ($T = 500$) is reached. Rendering this two-stage objective Markovian requires a hidden state $h_t$ that tracks the number of timesteps spent on the pad so far. Formally, reward is given by

$$R(s_t, a_t, h_{t+1}) = \begin{cases} R_{\text{pad}}(s_t) + R_{\text{shaping}}(s_t, 0) & \text{if } h_{t+1} < 50, \\ R_{\text{no\_contact}}(s_t) + R_{\text{hover}}(s_t) + R_{\text{shaping}}(s_t, 1) & \text{otherwise,} \end{cases}$$

where $R_{\text{pad}}$ rewards the agent for being central with both legs on the ground (i.e., on the pad),

$$R_{\text{pad}}([x_t, y_t, v_t^x, v_t^y, \theta_t, \dot{\theta}_t, c_t^l, c_t^r]) = \begin{cases} 1 & \text{if } -0.2 \leq x_t \leq 0.2 \text{ and } c_t^l = 1 \text{ and } c_t^r = 1, \\ 0 & \text{otherwise,} \end{cases}$$

$R_{\text{no\_contact}}$ rewards breaking leg-ground contact,

$$R_{\text{no\_contact}}([x_t, y_t, v_t^x, v_t^y, \theta_t, \dot{\theta}_t, c_t^l, c_t^r]) = \begin{cases} 1 & \text{if } c_t^l = 0 \text{ and } c_t^r = 0, \\ 0 & \text{otherwise,} \end{cases}$$

---

[6]Note the timestep indices used here, which result from the order in which environment states, hidden states and rewards are computed. At time $t$, the hidden state $h_t$ is first updated to $h_{t+1}$ by $\delta(h_t, s_t, a_t)$, then the reward is computed as $R(s_t, a_t, h_{t+1})$, and finally the environment state is updated to $s_{t+1}$ by $D(s_t, a_t)$.

$R_{\text{hover}}$ rewards aerial positions in a target zone above the pad,

$$R_{\text{hover}}([x_t, y_t, v_t^x, v_t^y, \theta_t, \dot{\theta}_t, c_t^l, c_t^r]) = \begin{cases} 1 & \text{if } -0.5 \leq x_t \leq 0.5 \text{ and } 0.75 \leq y_t \leq 1.25, \\ 0 & \text{otherwise,} \end{cases}$$

and $R_{\text{shaping}}$ promotes slow, stable, central flight towards a target vertical position $y_{\text{target}}$,

$$R_{\text{shaping}}([x_t, y_t, v_t^x, v_t^y, \theta_t, \dot{\theta}_t, c_t^l, c_t^r], y_{\text{target}}) =$$
$$0.1 \times \max\left(2 - \left(\sqrt{(x_t)^2 + (y_t - y_{\text{target}})^2} + \sqrt{(v_t^x)^2 + (v_t^y)^2} + |\theta_t| + |\dot{\theta}_t|\right), 0\right).$$

The hidden state dynamics are

$$h_{t+1} = \begin{cases} \min(h_t + 1, 50) & \text{if } R_{\text{pad}}(s_t) = 1, \\ h_t & \text{otherwise,} \end{cases}$$

with $h_0 = 0$.

## E.2 MIL Dataset Generation

We obtain datasets of several thousand trajectories per task, containing a wide distribution of outcomes and return values, as follows. For each task $k$, we define a discrete *trajectory classifier* function $C_k : \Xi \to \mathcal{C}_k$ and a limit $p_k$ on the proportion of trajectories in the dataset that are allowed to map to each class in $\mathcal{C}_k$. These are given as follows:

- **Timer:** $\mathcal{C}_{\text{timer}} = \text{num\_neg} \times \text{num\_pos}$, where $\text{num\_neg} = \{0..50\}$ counts the number of timesteps the agent spends in the treasure while its reward is negative ($t \leq 50$), and $\text{num\_pos} = \{0..50\}$ counts the number while the reward is positive ($t > 50$). The number of classes is $|\mathcal{C}_{\text{timer}}| = 51^2 = 2601$ and the per-class limit is $p_{\text{timer}} = 0.002$.

- **Moving:** $\mathcal{C}_{\text{moving}} = \text{num\_treasure}$, where $\text{num\_treasure} = \{0..100\}$ counts the timesteps spent in the treasure. $|\mathcal{C}_{\text{moving}}| = 101$ and $p_{\text{moving}} = 0.05$.

- **Key:** $\mathcal{C}_{\text{key}} = \{\text{no\_key}, \text{key\_no\_treasure}, \text{treasure}\}$, where the class is no\_key if the key is not collected, key\_no\_treasure if the key is collected but the treasure is not reached, and treasure if the treasure is reached after collecting the key. $|\mathcal{C}_{\text{key}}| = 3$ and $p_{\text{key}}$ is defined on a per-class basis: $0.25$ for no\_key and key, and $0.5$ for treasure.

- **Charger:** $\mathcal{C}_{\text{charger}} = \text{num\_treasure} \times \text{charge\_bin}$, where $\text{num\_treasure} = \{0..100\}$ counts the timesteps spent in the treasure and $\text{charge\_bin} = \{1..20\}$ is a binned representation of the *mean* charge level when in the treasure (e.g., 0.0 maps to bin 1, 0.48 to bin 10, 0.96 to bin 20). $|\mathcal{C}_{\text{charger}}| = 2020$ and $p_{\text{charger}} = 0.002$.

- **Lunar Lander:** $\mathcal{C}_{\text{lunar}} = \text{pad\_bin} \times \text{take\_off} \times \text{hover\_bin}$, where $\text{pad\_bin} = \{0, \{1..49\}, 50+\}$ is a binned representation of the number of timesteps spent on the landing pad (i.e., zero, fewer than 50 or at least 50), $\text{take\_off} = \{0, 1\}$ is a binary indicator of whether the lander takes off again after being on the pad, and $\text{hover\_bin} = \{0, \{1..19\}, 20+\}$ is a binned representation of the number of timesteps spent in the hover zone after being on the pad.[7] $|\mathcal{C}_{\text{lunar}}| = 18$, of which 9 are actually realisable (e.g., the lander cannot take off from the pad if it never reached it in the first place) and $p_{\text{lunar}} = 0.2$.

For $k \in \{\text{timer}, \text{moving}, \text{key}, \text{charger}\}$, a dataset $\mathbf{X}_k$, is assembled iteratively. On each iteration, we generate a length-100 trajectory, $\xi$, by sampling agent actions uniform-randomly from the action space (up, down, left, right, no-op) and running them through the simulator. Once the trajectory is complete, we evaluate its class $C_k(\xi)$. If there are already at least $p_k \times 5000$ trajectories in $\mathbf{X}_k$ with this class, $\xi$ is discarded. Otherwise, it is added to $\mathbf{X}_k$. This process repeats until $|\mathbf{X}_k| = 5000$.

The state-action space for Lunar Lander is too large for a random generate-and-select algorithm to terminate in any reasonable time. Instead, we recycle the length-500 trajectories generated as a by-product of training the oracle-based RL baselines (black curves in Figure 5, plus six more runs not included in the figure). Starting from a bank of 12000 trajectories, filtering based on the

---

[7]If the lander reaches the target of 50 timesteps on the pad, the time in the hover zone is measured from this point onwards. Otherwise, it is measured from the first timestep that the lander leaves the pad.

threshold $p_{\text{lunar}} = 0.2$ yields a final dataset $\mathbf{X}_{\text{lunar}}$ with 9762 trajectories. Although this approach of relying on oracle-trained agents to generate data may initially appear to "put the cart before the horse", we suggest that it provides a valuable test of the ability of our MIL models to learn from goal-directed (c.f. random) trajectories, and thus is a step closer to the online bootstrapping approach of simultaneous RM and RL, which we aim to tackle in future work (see Section 5.3).

### E.3   MIL Models and Hyperparameters

The MIL model training on the Timer, Moving, Key, and Charger tasks used the same process as for the toy model training (see Section D.2). However, we also applied dropout (DO) in these models. We give the MIL training hyperparameters for each of these tasks in Table A12. Again, the hyperparameters were found through a small amount of trial and error, i.e., no formal hyperparameter tuning was carried out.

Table A7: Timer, Moving, Key, and Charger task training hyperparameters.

| Dataset | LR | WD | DO | Patience | Epochs |
|---|---|---|---|---|---|
| Timer | $5 \times 10^{-4}$ | 0 | 0.1 | 50 | 250 |
| Moving | $5 \times 10^{-4}$ | 0 | 0.1 | 50 | 250 |
| Key | $5 \times 10^{-4}$ | 0 | 0.1 | 30 | 150 |
| Charger | $5 \times 10^{-4}$ | 0 | 0.1 | 50 | 250 |

In Tables A8 to A11 we give the architectures for the MIL models we used in the Timer, Moving, Key, and Charger tasks. As in the toy dataset experiments, the models are a combination of fully connected layers (FC) along with different MIL pooling mechanisms. Rectified linear unit (ReLU) activation is applied to the FC hidden layers. Again, we label the layers based on the part of the network they belong to: feature extractor (FE), head network (HN), or pooling (P); see Section 3.2. We also indicate the input and output sizes: $b$ x $n$ indicates an input or output where there is a representation of length $n$ for each of the $b$ instances ($b$ is the size of the input bag). Input features were normalised using mean/standard deviation scaling.

Table A8: RL Instance Space NN

| Layer | Type | Input | Output |
|---|---|---|---|
| 1 (FE-1) | FC + ReLU + DO | $b$ x 2 | $b$ x 64 |
| 2 (FE-2) | FC + ReLU + DO | $b$ x 64 | $b$ x 32 |
| 3 (FE-3) | FC + ReLU + DO | $b$ x 32 | $b$ x 32 |
| 4 (HN-1) | FC + ReLU + DO | $b$ x 32 | $b$ x 32 |
| 5 (HN-2) | FC + ReLU + DO | $b$ x 32 | $b$ x 16 |
| 6 (HN-3) | FC | $b$ x 16 | $b$ x 1 |
| 7 (P) | mil-sum | $b$ x 1 | 1 |

Table A9: RL Embedding Space LSTM

| Layer | Type | Input | Output |
|---|---|---|---|
| 1 (FE-1) | FC + ReLU + DO | $b$ x 2 | $b$ x 64 |
| 2 (FE-2) | FC + ReLU + DO | $b$ x 64 | $b$ x 32 |
| 3 (FE-3) | FC + ReLU + DO | $b$ x 32 | $b$ x 32 |
| 4 (P) | mil-emb-lstm | $b$ x 32 | 2 |
| 5 (HN-1) | FC + ReLU + DO | 2 | 32 |
| 6 (HN-2) | FC + ReLU + DO | 32 | 16 |
| 7 (HN-3) | FC | 16 | 1 |

Table A10: RL Instance Space LSTM

| Layer | Type | Input | Output |
|---|---|---|---|
| 1 (FE-1) | FC + ReLU + DO | $b$ x 2 | $b$ x 64 |
| 2 (FE-2) | FC + ReLU + DO | $b$ x 64 | $b$ x 32 |
| 3 (FE-3) | FC + ReLU + DO | $b$ x 32 | $b$ x 32 |
| 4 (P-1) | mil-ins-lstm | $b$ x 32 | $b$ x 2 |
| 5 (HN-1) | FC + ReLU + DO | $b$ x 2 | $b$ x 32 |
| 6 (HN-2) | FC + ReLU + DO | $b$ x 32 | $b$ x 16 |
| 7 (HN-3) | FC | $b$ x 16 | $b$ x 1 |
| 8 (P-2) | mil-sum | $b$ x 1 | 1 |

Table A11: RL CSC Instance Space LSTM

| Layer | Type | Input | Output |
|---|---|---|---|
| 1 (FE-1) | FC + ReLU + DO | $b$ x 2 | $b$ x 64 |
| 2 (FE-2) | FC + ReLU + DO | $b$ x 64 | $b$ x 32 |
| 3 (FE-3) | FC + ReLU + DO | $b$ x 32 | $b$ x 32 |
| 4 (P-1) | mil-csc-ins-lstm | $b$ x 32 | $b$ x 2 |
| 5 (HN-1) | FC + ReLU + DO | $b$ x 34 | $b$ x 32 |
| 6 (HN-2) | FC + ReLU + DO | $b$ x 32 | $b$ x 16 |
| 7 (HN-3) | FC | $b$ x 16 | $b$ x 1 |
| 8 (P-2) | mil-sum | $b$ x 1 | 1 |

### E.4 Lunar Lander MIL Models and Hyperparameters

There are several differences between the MIL training process for the Lunar Lander task and the other four RL tasks (see Appendix E.3). Firstly, the reward targets (and as such, return targets) were scaled down by a factor of 100 in order to avoid extremely large gradients from high prediction targets. For example, a trajectory with an original return of 700 would have a scaled return of 7. Secondly, the input data was scaled linearly between -0.5 and 0.5 (using the minimum and maximum range of each feature). This was found to give more consistent feature ranges than mean/standard deviation scaling as was used in the other tasks (this was due to large outliers in certain features, e.g., the rotational features were largely clustered around 0, but had extreme values up to $\pm$ 90). We give the training hyperparameters for the lunar lander environment in Table A12. Again, no formal hyperparameter tuning was carried out, so better performance of these models could potentially be achieved with better parameters (including shorter training times with a higher learning rate).

Table A12: Lunar Lander MIL training hyperparameters.

| Dataset | LR | WD | DO | Patience | Epochs |
|---|---|---|---|---|---|
| Lunar Lander | $1 \times 10^{-4}$ | 0 | 0 | 30 | 200 |

We used similar architectures to the other four RL tasks (see Appendix E.3), but with larger layers; see Tables A13 through A16. However, the depth of the models remained the same, as did the size of the hidden state embedding. Also note the addition of the Leaky ReLU activation function in the head networks, which we discuss further in Appendix E.5.

Table A13: Lunar Lander Instance Space NN

| Layer | Type | Input | Output |
|---|---|---|---|
| 1 (FE-1) | FC + ReLU + DO | $b$ x 2 | $b$ x 128 |
| 2 (FE-2) | FC + ReLU + DO | $b$ x 128 | $b$ x 64 |
| 3 (FE-3) | FC + ReLU + DO | $b$ x 64 | $b$ x 64 |
| 4 (HN-1) | FC + ReLU + DO | $b$ x 64 | $b$ x 64 |
| 5 (HN-2) | FC + ReLU + DO | $b$ x 64 | $b$ x 32 |
| 6 (HN-3) | FC + Leaky ReLU | $b$ x 32 | $b$ x 1 |
| 7 (P) | mil-sum | $b$ x 1 | 1 |

Table A14: Lunar Lander Emb. Space LSTM

| Layer | Type | Input | Output |
|---|---|---|---|
| 1 (FE-1) | FC + ReLU + DO | $b$ x 2 | $b$ x 128 |
| 2 (FE-2) | FC + ReLU + DO | $b$ x 128 | $b$ x 64 |
| 3 (FE-3) | FC + ReLU + DO | $b$ x 64 | $b$ x 64 |
| 4 (P) | mil-emb-lstm | $b$ x 64 | 2 |
| 5 (HN-1) | FC + ReLU + DO | 2 | 64 |
| 6 (HN-2) | FC + ReLU + DO | 64 | 32 |
| 7 (HN-3) | FC + Leaky ReLU | 32 | 1 |

Table A15: Lunar Lander Ins. Space LSTM

| Layer | Type | Input | Output |
|---|---|---|---|
| 1 (FE-1) | FC + ReLU + DO | $b$ x 2 | $b$ x 128 |
| 2 (FE-2) | FC + ReLU + DO | $b$ x 128 | $b$ x 64 |
| 3 (FE-3) | FC + ReLU + DO | $b$ x 64 | $b$ x 64 |
| 4 (P-1) | mil-ins-lstm | $b$ x 64 | $b$ x 2 |
| 5 (HN-1) | FC + ReLU + DO | $b$ x 2 | $b$ x 64 |
| 6 (HN-2) | FC + ReLU + DO | $b$ x 64 | $b$ x 32 |
| 7 (HN-3) | FC + Leaky ReLU | $b$ x 32 | $b$ x 1 |
| 8 (P-2) | mil-sum | $b$ x 1 | 1 |

Table A16: Lunar Lander CSC Ins. Space LSTM

| Layer | Type | Input | Output |
|---|---|---|---|
| 1 (FE-1) | FC + ReLU + DO | $b$ x 2 | $b$ x 128 |
| 2 (FE-2) | FC + ReLU + DO | $b$ x 128 | $b$ x 64 |
| 3 (FE-3) | FC + ReLU + DO | $b$ x 64 | $b$ x 64 |
| 4 (P-1) | mil-csc-ins-lstm | $b$ x 64 | $b$ x 2 |
| 5 (HN-1) | FC + ReLU + DO | $b$ x 66 | $b$ x 64 |
| 6 (HN-2) | FC + ReLU + DO | $b$ x 64 | $b$ x 32 |
| 7 (HN-3) | FC + Leaky ReLU | $b$ x 32 | $b$ x 1 |
| 8 (P-2) | mil-sum | $b$ x 1 | 1 |

### E.5 Lunar Lander MIL Discussion

In Table 1, we present results for Lunar Lander using only the top five performing models by reward MSE (50% of all models). In this section, we discuss why this is the case.

When training the Lunar Lander architectures with linear activation functions in the head networks, we found that the models were struggling to learn the correct return predictions as they were making negative reward predictions. We know *a priori* that the Lunar Lander task only has positive rewards, therefore we added a Leaky ReLU activation (with a negative slope of $1 \times 10^{-6}$) to each architecture's head network to encourage positive predictions. This led to an immediate improvement

in performance for all model architecture (excluding the Instance Space NN baseline, but this was expected to fail on this task as it cannot model temporal dependencies).

However, we found that a proportion of model initialisations remained unable to overcome a certain local minimum during training (corresponding to a return prediction MSE of around 2.1). In other cases, after this threshold was passed, the model performance would rapidly improve. Note this problem occurred in each of the LSTM-based models (the Instance Space NN architecture was never observed to pass this threshold). Therefore, we focus our evaluation on the top 50% of models, which is equivalent to discarding the worse-performing models, the majority of which had not passed the problematic threshold during training.

We investigate this training issue further in Figure A1, focusing specifically on the CSC Instance Space LSTM models. Although the Leaky ReLU activation encourages models to make positive predictions, we can see that it does not prevent them entirely. Furthermore, the models that are not able to cross the return threshold of 2.1 tend to output a greater proportion of negative reward predictions. We thus hypothesise that a better mechanism for preventing negative reward prediction would increase the chance that a given model training run achieves a return loss of less than 2.1. Below we list several such mechanisms as alternatives to our current Leaky ReLU approach.

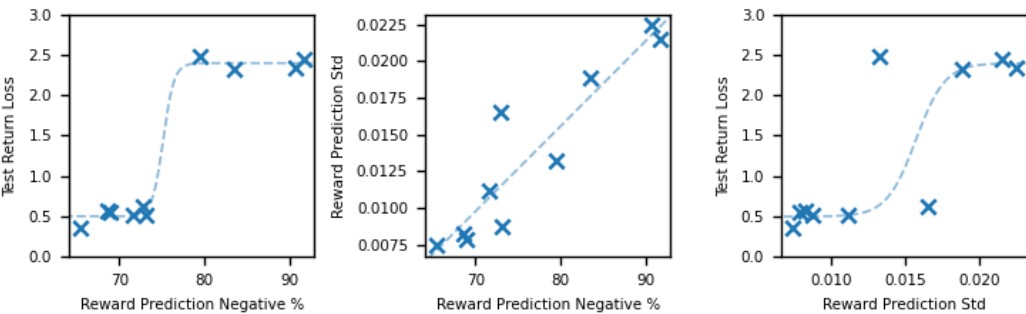

Figure A1: An analysis of the negative reward prediction of the ten Lunar Lander CSC Instance Space LSTM models trained in this work. Note we observed similar trends for the Embedding Space LSTM and Instance Space LSTM models. **Left:** Models that are able to cross the return loss threshold of 2.1 make fewer negative reward predictions (given here as a percentage of all reward predictions) than those that cannot. **Middle:** As the number of negative reward predictions increases, so too does the standard deviation of the reward predictions. In order to sum to the correct return predictions, the positive reward predictions must increase to compensate for the negative reward predictions, leading to a larger variance, and ultimately worse reward prediction. **Right:** The standard deviation of the reward predictions also highlights the loss threshold of 2.1, although not as clearly as the percentage of negative reward predictions.

**Replace Leaky ReLU with ReLU** One option to remove negative reward predictions entirely is to use ReLU rather than Leaky ReLU in the final activation function of the head networks. However, in the case that all the reward predictions are negative, the gradient of the network will be zero, so no learning can take place. The ability of the network to learn is entirely dependent on achieving at least one positive prediction in order to generate non-zero gradients, which is determined by the initial network weights. This could potentially be overcome with different network initialisation approaches, or by simply discarding training runs that fail to begin to learn.

**Replace Leaky ReLU with Sigmoid** Instead of using a Leaky ReLU or ReLU activation in the head network, the Sigmoid activation function could be used. This would overcome the gradient issue presented with the use of ReLU, and would ensure that no negative rewards are predicted. However, this would require *a priori* knowledge of the maximum reward target in order to scale the network outputs correctly, and the non-linear activation could make accurate prediction of rewards more difficult in some cases.

**Remove target normalisation** A further approach to reduce the number of negative reward predictions would be to remove, or at least reduce, the reward target scaling. As discussed above, this was initially included to avoid very large gradients. However, reducing this scaling would move the

average reward prediction away from zero, potentially reducing the number of negative predictions. Care would have to be taken to not reintroduce the large gradient problem.

**Regularise reward prediction variance** A final alternative approach could be to introduce an additional loss term that encourages the reward predictions to have low variance. This would deter a large range of reward predictions, leading indirectly to fewer negative reward predictions (see Figure A1). However, this approach is only applicable to the LSTM models that produce instance predictions, i.e., it cannot be used with the Embedding Space LSTM network as that architecture does not produce reward predictions during training. Furthermore, this would result in an additional hyperparameter that requires tuning (a coefficient for the new loss term).

## F   Further Details on RL Agent Training

Adopting OpenAI Gym terminology [9], non-Markovian reward functions (both ground truth oracles and learnt LSTM-based models) are implemented as *wrappers* on rewardless base environments. The role of a wrapper is to track the hidden state of either the oracle or the LSTM throughout an episode and use this to compute rewards to return to the agent. In the "Oracle (without hidden state)" baseline, we return the raw environment state (e.g., the 2D position $[x_t, y_t]$) to the agent unmodified. Otherwise, we concatenate the post-update hidden state $h_{t+1}$ onto the end of the environment state, thereby expanding the state space from the agent's perspective and making rewards Markovian.

This wrapper-based approach allows us to use a completely vanilla RL algorithm. For the 2D navigation tasks, we use a Deep Q-Network (DQN) agent [31] with the double Q-learning trick [43] enabled. For Lunar Lander, which has a continuous action space, we use Soft Actor-Critic (SAC) [15]. In both cases we use a value network with ReLU activation functions, which is updated on every timestep by sampling batches of size 128 from a replay buffer. Bellman updates use a discount factor of $\gamma = 0.99$ and are implemented by the Adam optimiser with a learning rate of $1e^{-3}$. A target network tracks the primary one by Polyak averaging of parameters with a coefficient of $5e^{-3}$ per timestep. Additional hyperparameters are given below:

- **2D tasks (DQN):** number of training episodes $= 400$; replay buffer capacity $= 5e^4$; value network hidden layer sizes $= [256, 128, 64]$; policy greediness $\epsilon$ linearly decayed from 1 to 0.05 over the first 200 episodes and held constant thereafter.
- **Lunar Lander (SAC):** number of training episodes $= 800$; replay buffer capacity $= 1e^5$; value/policy network hidden layer sizes $= [256, 256]$; policy entropy regularisation coefficient $\alpha = 0.2$; policy updates with Adam optimiser (learning rate $1e^{-4}$).

## G   Trajectory Probing for Other RL Tasks

In this section, we provide additional trajectory probing plots (like Figure 9) for the Timer (Figure A2), Moving (Figure A3), Key (Figure A4), and Lunar Lander (Figure A5) tasks. As before, in these probing plots, we analyse the best-performing CSC Instance Space LSTM model for each task (according to the reward reconstruction metric).

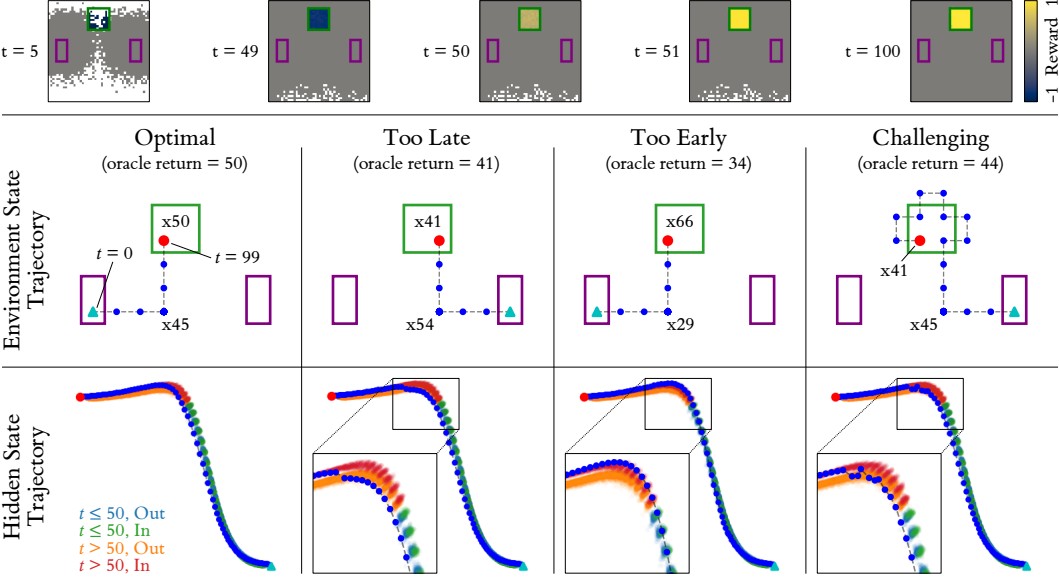

Figure A2: Timer task trajectory probing.
**Top**: Predicted reward with respect to time and position. We observe that the model has correctly captured the transition from negative to positive reward in the treasure region at $t = 50$, with no reward outside of this region. Although the reward is positive at $t = 50$, the model is uncertain at this point, i.e., the transition from negative to positive reward happens over two timesteps rather than one.
**Middle/bottom**: Four trajectory probes demonstrating the model's hidden state transitions. "x$n$" labels indicate the agent remaining in a position for $n$ timesteps.
*Optimal*: The agent moves into the treasure region at $t = 50$ and remains there, receiving the maximum possible reward.
*Too Late*: The agent moves into the treasure region a while after the treasure has already become positive, i.e., it is missing out on reward by not being in the region for as long as possible. This is reflected in the hidden state plot, where the state transitions from the orange to the red region after the $t = 50$ boundary.
*Too Early*: The agent moves into the treasure region before the treasure becomes positive, therefore, while it earns the maximum amount of positive reward, it also earns negative reward, leading to a sub-optimal result.
*Challenging*: The agent moves into the treasure region at the correct time, but proceeds to jump in and out of the treasure region before settling, leading to lost reward. The hidden state trajectories somewhat mimic this movement by transitioning between the orange and red regions, although the jumps are less clear near to the $t = 50$ transition point, suggesting the model is uncertain at this point.

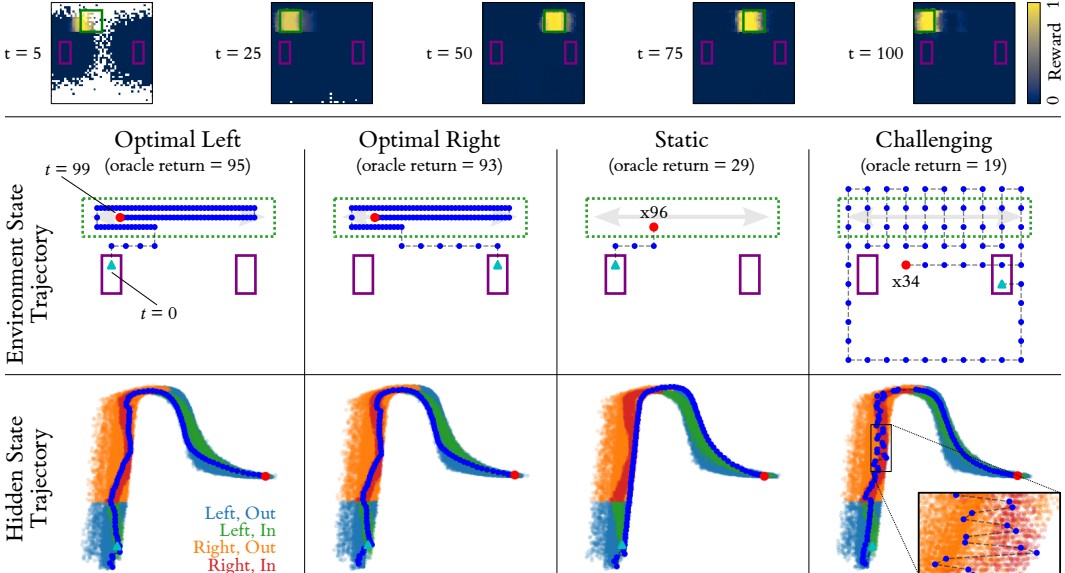

Figure A3: Moving task trajectory probing.

**Top**: The predicted reward with respect to time (and thus implicitly, the position of the treasure region). The model has learnt to track the treasure region as it moves, although there is noise around the left and right edges of the region, highlighting the difficulty of recovering the treasure region's horizontal position exactly.

**Middle/bottom**: Four trajectory probes showing the model's hidden state transitions. Note the green dotted region indicates the overall boundary of the treasure region, i.e., the treasure lies somewhere within that boundary, with its true horizontal position dependent on time. "x$n$" labels indicate the agent remaining in a position for $n$ timesteps.

*Optimal Left*: The agent moves within the treasure region as quickly as possible and then moves with the treasure (keeping within it) for the remainder of the episode. The hidden state trajectories follow the "In" regions (green and red).

*Optimal Right*: A similar optimal probe where the agent starts from the right-hand spawn zone rather than the left. In this case, the agent has further to move before moving into the treasure region, leading to slightly less overall reward than when it starts from the left.

*Static*: Rather than moving with the treasure region, the agent stays still and allows the treasure to pass over it, gaining reward for some timesteps but not others. We observe that the hidden state trajectory also reflects this — the state transitions between the "In" and "Out" regions.

*Challenging*: The agent takes a while to move towards the treasure region, and then passes in and out of the boundary in which the treasure resides, picking up some reward. We can see from the hidden state trajectory that this motion is captured in the hidden states — the trajectory transitions between the orange and red regions as the agent passes back and forth through the treasure region.

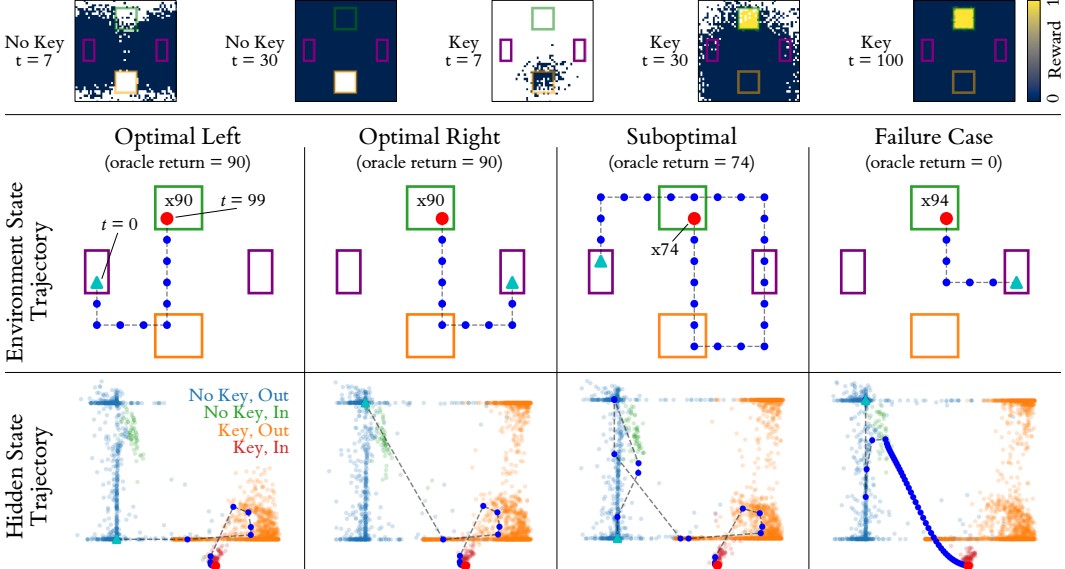

Figure A4: Key task trajectory probing.

**Top**: The predicted reward with respect to time and collection of the key. The model has learnt to only give reward when the agent is in the treasure region after collecting the key.

**Middle/bottom**: Four trajectory probes showing the model's hidden state transitions. "x$n$" labels indicate the agent remaining in a position for $n$ timesteps.

*Optimal Left*: The agent collects the key as quickly as possible, and then proceeds to move into the treasure region and wait there until the end of the episode. The hidden state trajectory shows two notable transitions, first when the key is collected (blue to orange) and then when the agent enters the treasure region (orange to red).

*Optimal Right*: A similar optimal policy, but from the right-hand spawn zone rather than the left. Note the hidden state trajectory is very similar, apart from the initial hidden state, which is at the opposite side of the blue region, representing the different spawn position.

*Suboptimal*: The agent passes through the treasure region before collecting the key, and then follows an optimal policy. We observe a transition in the hidden state trajectory from the blue to the green region that corresponds to the agent's premature entrance into the treasure region.

*Failure case*: The agent enters the treasure region without collecting the key and remains there for the rest of the episode. This highlights a failure case of the model, where the hidden state "drifts" from the green region to the red region, i.e., the model convinces itself that the agent must have picked up the key at some point. Such *causal confusion* errors are well-documented in the RM literature [40]. In our case, we suspect that the error is due to a bias in our data generation method inducing a correlation between key possession and time spent in the treasure region. As described in Appendix E.2, we specifically screen for trajectories where the treasure is visited with the key (the "treasure" class) but not for those where it is visited without it. There are thus likely to be very few training trajectories that spend a lot of time in the treasure region without collecting the key, making this probe an extreme outlier on which performance is poor. Adding and selecting for a fourth "key_no_treasure" class during data generation may have mitigated this issue.

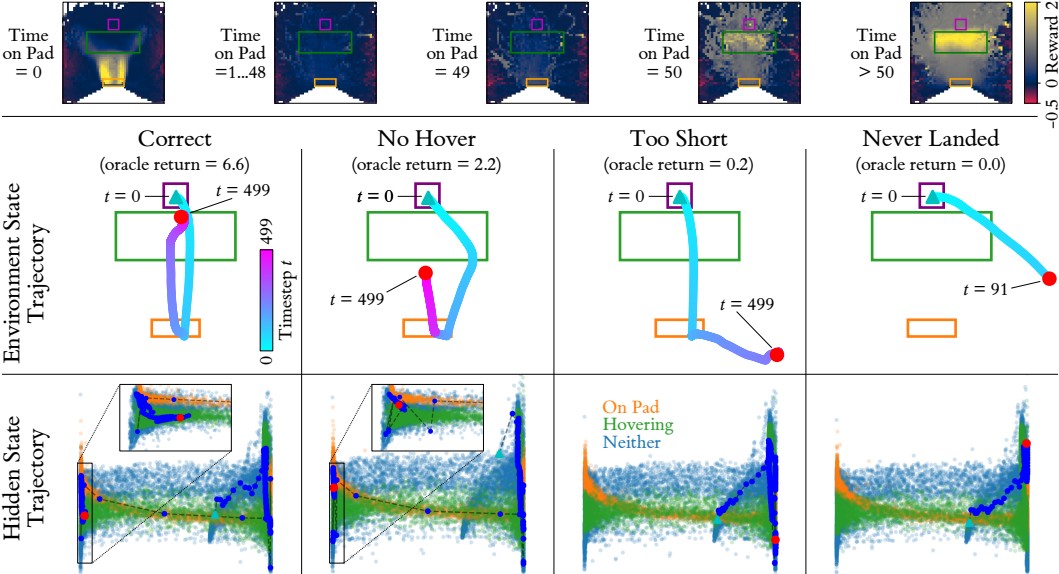

Figure A5: Lunar Lander task trajectory probing.

**Top**: The predicted reward with respect to the agent's position and the length of time that it has been on the pad. The model has learnt to reward the agent for landing on the pad when it has not previously landed, and then reward it for taking off and remaining in the hover zone after 50 timesteps on the pad, as per the oracle. However, observe how the top half of the hover zone is attributed higher reward than the lower half. This feature is not present in the oracle (which rewards the entire hover zone equally) but makes great practical sense, as a stochastic policy is less likely to drop out of the zone under the effect of gravity if it remains some distance from the bottom edge. Also note the small negative rewards near the environment boundaries, which disincentivise positions with a high risk of leading to early termination, and the "funnel" of intermediate positive reward, which may help to guide the agent up from the pad to the hover zone. Collectively, these deviations from the oracle reward function could actually be seen as *improvements*, and explain why we observe significantly higher performance on the $R_{\text{hover}}$ reward component when the CSC Instance Space LSTM reward model is used, compared with using the oracle itself.

**Middle/bottom**: Four trajectory probes showing the model's hidden state transitions.

*Correct*: The agent lands on the pad as quickly as possible, waits for 50 timesteps, and then takes off again and hovers in the hover zone. The hidden state trajectory shows a transition from right to left once the agent has been on the pad for long enough. It also clearly shows the agent is in the Hovering hidden state region.

*No Hover*: In this trajectory, the agent remains on the pad for too long and does not enter the hover zone. It has a similar hidden state transition to the *Correct* trajectory but does not enter the Hovering hidden state region.

*Too Short*: The agent lands on the pad but does not stay there for the required 50 timesteps. In this particular case, the agent lands correctly but then slips out of the landing pad, coming to rest on a different area of terrain until the end of the trajectory. In the hidden state trajectory, there is no transition from the right side to the left side, matching the idea that the agent has not remained on the pad for long enough.

*Never Landed*: The agent does not land on the pad at all. In this particular example, the agent reaches the boundary of the environment, which causes the episode to terminate early. Similar to the *Too Short* example, the hidden state trajectory does not transition from right to left.