# OpenReview forum: "Non-Markovian Reward Modelling from Trajectory Labels via Interpretable Multiple Instance Learning"
_NeurIPS.cc/2022/Conference — NeurIPS 2022 Accept_

### Official Review · Reviewer_kcLW · 2022-07-11

**Rating:** 6
**Confidence:** 4
**Soundness:** 3 good
**Presentation:** 4 excellent
**Contribution:** 3 good

**Summary:**

This paper defines non-Markovian reward modeling and proposes a novel LSTM-based model to capture temporal dependencies in reward modeling. It also adopts a multiple instance learning (MIL) framework to handle temporal dependencies with reward labels. It encodes hidden state information using LSTM and also produces explicit reward predictions with skip-connection to the current state and action features.
It optimizes LSTM-based models on offline trajectory datasets. Quantitative experimental results on return and reward prediction show the model's compelling reward function modeling quality. The authors also present qualitative results on the interpretability of the learned hidden embeddings.


**Questions:**

1. How important the proposed non-Markovian reward modeling is compared to other temporal information encoding models (such as Flare [1] or SPR [2])?

2. How many timesteps the proposed LSTM-based model can encode?

3. What did you use for the LSTM hidden states 2D visualization? T-SNE? And do you have any other visualization results on other common RL tasks with discrete action space such as Atari games?


- [1] Reinforcement Learning with Latent Flow, Shang et. al., NeurIPS 2021
- [2] Data-Efficient Reinforcement Learning with Self-Predictive Representations, Schwarzer et. al., ICLR 2021

**Limitations:**

Yes, the authors adequately addressed the limitations and potential negative social impact of their work.

**Strengths And Weaknesses:**

Strengths:

The paper is well written and easy to read and understand. Figure 2 clearly demonstrates the proposed method and the baseline models. The experiments are comprehensive and the visualization result is quite inspiring. They also propose novel RL tasks that can show the effectiveness of their model. For significance, learning the non-Markovian reward function is the cornerstone of a successful real-world application using RL, especially with human interactions.

Weaknesses:

I appreciate the idea and results of the conceptual toy RL tasks experiments, however, it would be better if there were comparisons with other models in the more complex RL tasks such as Montezuma's Revenge where modeling non-Markovian rewards is important.

---

> ### Author Response · Authors · 2022-08-02
> **Author Response to Reviewer kcLW**
>
> **Q1.** *It would be better if there were comparisons with other models in the more complex RL tasks such as Montezuma's Revenge where modeling non-Markovian rewards is important.*
>
> **A1.** Please see our discussion of more complex environments in our general rebuttal. The reason we steered clear of Atari environments with visual observations was partly due to complexity, but primarily due to the fact that these are well known to be partially observable, so have non-Markovian *dynamics* (not just rewards) with respect to their observations. In our view, untangling the two modes of departure from the Markovian case (partial observability and non-Markovian task specifications) would have led to a messy and needlessly complicated paper whose core message may have been lost. We should note, though, that our models could easily be combined with a CNN that maps (stacked) image observations into environment state vector, so could be deployed on Atari games in future.
>
> **Q2.** *How important the proposed non-Markovian reward modeling is compared to other temporal information encoding models (such as Flare or SPR)?*
>
> **A2.** Although we are not intimately familiar with the particular models you mention, they both appear to be specialised methods for improving RL from partial (pixel) observations. Flare aims to learn a compact representation of the smooth evolution of a dynamical system, while SPR aims to learn a representation that is "self-predictive" of its own state a small number of timesteps into the future. Hence, both are concerned with modelling the short-timescale dynamics of a general partially-observed state, and not for reconstructing a reward function with potentially long-term dependencies and no guarantee of smooth evolution over time. For the latter case, we believe that our LSTM-based approach is far more suitable.
>
> **Q3.** *How many timesteps the proposed LSTM-based model can encode?*
>
> **A3.** The most we have tested on so far is 100 timesteps per episode. This number comes from our RL tasks (Section 4.1), which used a fixed episode length of 100. LSTMs have no hard constraint on the number of sequential inputs that can be processed, although long-term dependencies become increasingly challenging to maintain without corruption. In our planned revised version of the paper (please see our general rebuttal), we will add a modified LunarLander environment from Open AI Gym with upwards of 400 timesteps per episode.
>
> **Q4.** *What did you use for the LSTM hidden states 2D visualization? T-SNE?*
>
> **A4.** We did not need to use any dimensionality reduction for the visualisation. As stated at the end of Section 4.1, "...we know a priori that it is possible to capture the temporal dependencies in at most two dimensions, therefore we constrain our models to use 2D hidden states." As the models used 2D hidden states, we were able to produce the plots directly from the hidden states without any sort of transformation. This was an intended part of the model design. However, we are aware that for more complex models/environments, hidden states with higher dimensionality may need to be used, and as such, a method like t-SNE would indeed be required to produce visualisation similar to the ones we produced in this work.

---

### Official Review · Reviewer_2DcX · 2022-07-11

**Rating:** 4
**Confidence:** 4
**Soundness:** 3 good
**Presentation:** 4 excellent
**Contribution:** 2 fair

**Summary:**

This paper proposes a new method to deal with non-Markovian rewards based on reformulating the problem as multiple instance learning with LSTM. The authors test a few alternative ways of doing this in 4 environments designed to demonstrate non-Markovain properties.

**Questions:**

- Could the authors clarify if the issue with the difference in partial bags as I described takes place in their experiments?
- Could the authors clarify why it is chosen for the agent to reuse latent states from reward modelling rather than to train its own LSTM?
- Could the authors elaborate why the baseline that just replaces each state with frame stacked state (several states concatenated, often used as a simplification for recurrence) would not suit this problem?
- The authors train Deep Q-Network, but as far as I understand the states are just 2-dimensional, how important is it to have deep architectures in this case?
- The authors say that the transformer architecture would be unsuitable for the temporal structure of the problem. Why would it be the case with temporally masked transformers?
- How was the dataset for RM training (from lines 210-213) split into training and test? Are the results in Table 1 on the test set?


**Limitations:**

Some limitations are discussed at the end of the paper. In the review I also pointed to/asked for clarification regarding some potential limitations. No potential societal impact is mentioned.

**Strengths And Weaknesses:**

Originality:

While Markovian reward modelling is quite well studied, non-Markovian rewards are investigated much less. The proposed improvements to MIL with LSTM (instance space and skip connections) are not particularly original, but they effectively address the shortcomings of the prior methods.

Quality:

I am not completely convinced about the baseline "Base Case: Embedding Space LSTM" model where the rewards are obtained by taking the difference in partial bag labels. Suppose the task is to grasp an object. If I am given two sequences to annotate: 1) s1=not grasped, s2=not grasped, s3=not grasped, s4=grasped, s5=grasped and 2) s1=not grasped, s2=not grasped, s3=not grasped, s4=grasped, I could assign both of them label 1. Then, using the difference between partial bags would result in the reward of s5 to be 0 which is intuitively not correct. I expect that with predictions from LSTM we may face a similar problem. If it is the case, maybe the inferior performance of this method could be due to this.
In section 3.3 the authors explain that the RL agent observes hidden states along the reward from LSTM. However, I am wondering if it is suitable to use hidden state representation of the reward model for agent training. Maybe it is better to have a recurrent agent architecture that trains its own latent representation.
For the baselines in Figure 5, I think it would be more informative to have RL with recurrent architecture + oracle rewards as the upper bound on the performance and it would avoid the situation when an agent with learnt reward outperforms the oracle.

Clarity:

The paper is very well written and easy to follow. I found the visualizations in Figures 7 and 8 very informative.
The paper positions its problem as originating from an unrealistic assumption on human evaluation of temporally-extended behaviour. However, I believe the non-Markovian rewards might be important in practice in some tasks that are inherently non-Markovian and the success can't be determined only by a given state, but a whole sequence must be considered. Some of such tasks are studied in the experiments, but the point that these are different types of tasks was not made clear. It could be also informative as an additional ablation to consider simple Markovian tasks (with Markovian reward), for example, like navigating to a given target to confirm that the method of this paper works in that case too.

Significance:

My biggest concern in this paper is the complexity of the evaluation environments. While the proposed environments and tasks work sufficiently well to demonstrate the proof of concept, they are rather simplistic (grid world) and some tasks are a bit artificial. I am wondering if the proposed method would face any difficulties when dealing with a more challenging environment, for example, some control problems in simulation, or something that has image-based states. I think that in order to have a larger impact in the community, this paper should include results on more complex environments.

---

> ### Author Response · Authors · 2022-08-02
> **Author Response to Reviewer 2DcX (1/2)**
>
> **Q1.** *I am not completely convinced about the baseline ``Base Case: Embedding Space LSTM" model where the rewards are obtained by taking the difference in partial bag labels. Suppose the task is to grasp an object. If I am given two sequences to annotate: 1) s1=not grasped, s2=not grasped, s3=not grasped, s4=grasped, s5=grasped and 2) s1=not grasped, s2=not grasped, s3=not grasped, s4=grasped, I could assign both of them label 1. Then, using the difference between partial bags would result in the reward of s5 to be 0 which is intuitively not correct. / Could the authors clarify if the issue with the difference in partial bags as I described takes place in their experiments?*
>
> **A1.** You are correct in saying s5 would be attributed a value of 0. However, this may not necessarily be incorrect; it depends on how a hypothetical human evaluator chooses (or is told) to score the model. If the agent's goal is just to *grasp the the object at some point*, then strictly rewarding only s4 is correct: is does not matter if the agent drops the ball later, so no further reward should be attributed to s5. If the goal is to *grasp the object and not let go*, for example with a score of 1 if the object is grasped and held, and 0 otherwise (never grasped or grasped and subsequently dropped), then one could envisage the reward breakdown being +1 for grasping and -1 for letting go, in which case only attributing s4 is again correct. If the goal is to *grasp the object for as long as possible*, then the scores assigned to the two scenarios should be different (e.g., 2 for the first example you give and 1 for the second). In summary, we see no issue here; the model can and should assign rewards in a way that is consistent with how the evaluator scores the trajectories.
>
> **Q2.** *Maybe it is better to have a recurrent agent architecture that trains its own latent representation. Could the authors clarify why it is chosen for the agent to reuse latent states from reward modelling rather than to train its own LSTM?*
>
> **A2.** Firstly, the question of whether the reward modelling process, and the learnt LSTM model, should be seen as situated inside the agent, as opposed to within some distinct software system, is one of arbitrary semantics, and you might find it helpful to reframe our entire reward modelling pipeline as an activity performed *by the agent* alongside its primary task of policy learning. A more fundamental distinction is between *offline* reward modelling, where the agent learns and fixes its reward model prior to commencing policy learning, and *online* reward modelling, where both the reward and policy models are learnt concurrently from streaming human feedback. We only investigated the former case here, but the latter is likely to be more common in practice. In section 5.3 of the paper, we identify experiments with the online case as a major area for future work.
>
> **Q3.** *For the baselines in Figure 5, I think it would be more informative to have RL with recurrent architecture + oracle rewards as the upper bound on the performance and it would avoid the situation when an agent with learnt reward outperforms the oracle.*
>
> **A3.** If we understand correctly, your suggestion is to baseline against an LSTM-based agent that is given oracle rewards but *not* oracle hidden states, as a kind of "halfway house" between direct oracle access and full reward modelling. From a technical perspective, this would have been an entirely reasonable thing to do, and our main response is simply that this would have added further complexity to some already-dense figures. We would also argue that such a halfway house doesn't really correspond to a realistic setup in the reward modelling context, where it is generally assumed that the oracle (or in reality, the human) cannot provide reward labels directly, but only high-level feedback such as trajectory labels. However, we will bear this suggested baseline in mind for future experiments.
>
> **Q4.** *I believe the non-Markovian rewards might be important in practice in some tasks that are inherently non-Markovian and the success can't be determined only by a given state, but a whole sequence must be considered. Some of such tasks are studied in the experiments, but the point that these are different types of tasks was not made clear.*
>
> **A4.** We provide a preliminary taxonomy of non-Markovian tasks in Appendix B; does this provide the kind of discussion you are looking for?

---

> > ### Author Response · Authors · 2022-08-02
> > **Author Response to Reviewer 2DcX (2/2)**
> >
> > **Q5.** *It could be also informative as an additional ablation to consider simple Markovian tasks (with Markovian reward).*
> >
> > **A5.** We do precisely this in our "Toggle Switch" toy dataset in Appendix C (C.3 for the results). We found that the baseline Instance Space NN did work on this task as expected, but was still outperformed by our CSC Instance Space LSTM for both return and reward prediction. Although this is only one example, it does suggest there is some potential for non-Markovian models to provide performance benefits even in the Markovian case, perhaps because they are able to learn a more flexible, reward-relevant representation of the environment than is available in the default (usually hand-engineered) state vector.
> >
> > **Q6.** *My biggest concern in this paper is the complexity of the evaluation environments. While the proposed environments and tasks work sufficiently well to demonstrate the proof of concept, they are rather simplistic (grid world)...*
> >
> > **A6.** Please see our comments in the general rebuttal with regards to using more complex environments and adding another experiment to a revised version of the paper. As a technical point, please note that the environments studied so far are not strictly grid worlds (which have discrete state spaces and are amenable to tabular RL algorithms) but 2D continuous environments that require function approximation to solve.
> >
> > **Q7.** *Could the authors elaborate why the baseline that just replaces each state with frame stacked state would not suit this problem?*
> >
> > **A7.** Frame stacking is a reasonable proxy for recurrence when a small number of timesteps (typically $3-5$) is sufficient to create an approximately Markovian representation. The long-term dependencies studied in this work do not satisfy this assumption, and would necessitate tens of timesteps of stacking, creating an unwieldy and inefficient representation. Furthermore, the Timer and Moving tasks fundamentally require the recovery of indexical information (i.e., the current timestep $t$) which no amount of stacking would provide. It might have been reasonable to consider frame stacking with $5$ timesteps as a low-quality baseline in our experiments, although we can be extremely confident that it would have been ineffective given the structure of the tasks.
> >
> > **Q8.** *The authors train Deep Q-Network, but as far as I understand the states are just 2-dimensional, how important is it to have deep architectures in this case?*
> >
> > **A8.** As mentioned above, the continuous nature of the state space means the agent must utilise some form of function approximation, regardless of the dimensionality. Older (pre-2013) work on RL with function approximation explored many options such as linear functions and radial basis functions, although these constrained function classes (especially linear) would likely perform poorly in the highly nonlinear and discontinuous world of "regions" and "thresholds" in our experimental environments. Since the 2013 DQN paper, neural network function approximation has come to dominate the literature, so we see our decision to follow this direction as entirely consistent with what most RL researchers would have done. As for the implication of a "deep" (vs shallow) network, we suggest that our architecture size is rather modest by modern standards, although we concede that further tuning may have found a yet smaller architecture that performs well on the tasks considered. Such tuning was not the focus of our work.
> >
> > **Q9.** *The authors say that the transformer architecture would be unsuitable for the temporal structure of the problem. Why would it be the case with temporally masked transformers?*
> >
> > **A9.** Could the reviewer please indicate a particular paper or example of a temporally masked transformer that they believe would be suitable? As a general note, we opted against transformers in this work because they lack an explicit analogue of a hidden state which is carried forward through time, which is the most parsimonious way of representing non-Markovian rewards. The hidden states recovered by our LSTM models also provide a meaningful object of study in our interpretability investigations; the propagation of dependencies within a transformer architecture is far less easy to visualise and interpret.
> >
> > **Q10.** *How was the dataset for RM training (from lines 210-213) split into training and test?*
> >
> > **A10.** As discussed in the Appendix (C.3 and D.2), we used an 80/10/10 dataset split (train/validation/test), which was controlled using fixed seeds.
> >
> > **Q11.** *Are the results in Table 1 on the test set?*
> >
> > **A11.** Yes, these are on the test set. We're happy to add that clarification to the table caption.

---

### Official Review · Reviewer_2P9E · 2022-07-12

**Rating:** 7
**Confidence:** 4
**Soundness:** 4 excellent
**Presentation:** 3 good
**Contribution:** 2 fair

**Summary:**

This paper presents reward modeling methods for non-markovian rewards, leveraging LSTMs to update a hidden state and encode the trajectory history. It performs experiments in custom environments with non-markovian rewards, with a baseline that assumes markovian rewards. Their various proposed methods perform better than the baseline. The theory behind the method is sound and the experiments are sound too.

**Questions:**

Without resorting to completely standard benchmarks where the reward is markovian, could you make a stronger case that the methods are applicable and helpful for larger-scale, real-world tasks?

And more importantly, can your method actually handle the complexity of human-labels rewards as you talk about in the introduction but don’t demonstrate in your experiments?

**Limitations:**

It is not clear what the limitation of this work is as it has to potential to be very general; the experiments do not go far enough to find the limits of the methods. I see no obvious negative societal impact of this work.

**Strengths And Weaknesses:**

Good points: I think the connection between reward modeling with trajectory-level labeling and multi-instance learning is interesting and well explained. The experiments give good (although artificial) examples of non-markovian reward decision process. The writing is good and the paper is clear, and the reviewer had a good understanding of the presented methods after reading the paper.

My main gripe with the method and theory is the use of the NMRDP (non-markovian reward decision process), which can be viewed as a specialisation of a POMDP where only the reward depend on the hidden state. This would be a useful formalism if it led to a simpler algorithm, but the described algorithm resembles exactly what you would get if you tried to solve the reward modeling problem for POMDPs using LSTMs. I realise that the experiments only contain the special case of NMRDPs, but there is a missed opportunity in presenting a more general solution here. The argument of using this formalism to model human behaviour when assigning rewards is unconvincing to me as it is not tested in the experiment.

Speaking of this, I feel the experiments are too weak and obviously make it impossible for the baseline to perform well. They do demonstrate that the presented methods can handle non-markovian rewards in these limited settings. The problem is the applicability of this setting to more complex or better-known tasks; without resorting to completely standard benchmarks where the reward is markovian, could you make a stronger case that the methods are applicable and helpful for larger-scale, real-world tasks? And more importantly, can your method actually handle the complexity of human-labels rewards as you talk about in the introduction but don’t demonstrate in your experiments? In my opinion, addressing the second question with a set of experiment with real human labels would also address the first question.

To summarize 1) I think the method would be better presented as reward-modeling for POMDPs (where lstms have been used to encode the history in many works, see Dreamer, PlaNet, reccurent DQN) at which point the solution becomes obvious. 2) The experiments, while valid, do not convince the reviewer of the wider applicability of their methods.

Small comments:

The abstract uses a lot of specialized language RM, MIL, “provide interpretable learn hidden information” that wasn’t really clear to the reviewer until *after* reading the paper and coming back to the abstract.

I find it awkward that the reward is a function of both s_t and h_{t+1}. I realize that it leads to better performance in practice, but it shouldn’t be necessary and maybe drop it from the math.

Section 3.2, and the paper in general, completely ignores the existing body of work using LSTMs (and RNNs in general) for POMDPs, which leads to algorithms similar to yours (see for example Hausknecht, Matthew, and Peter Stone. "Deep recurrent q-learning for partially observable mdps." 2015 aaai fall symposium series. 2015.) I think at least a mention should be added; I think the idea of using LSTMs specifically to handle non-markovian reward is simply a specialisation of this existing idea of using RNNs to encode past observations in POMDPs. It's not a bad thing per se, but should be mentioned.

Line 229 mention an “interesting” case when CSC and instance-space LSTM beats the oracles; the reviewer does not think it is interesting but rather suspicious. This could be stemming from a lack of hyperparameter search or uncertainty in the results.

---

> ### Author Response · Authors · 2022-08-02
> **Author Response to Reviewer 2P9E**
>
> Please refer to the general rebuttal for answers to your questions about using more complex environments and real human labels. Below are responses to other concerns raised:
>
> **Q1.** *My main gripe with the method and theory is the use of the NMRDP (non-markovian reward decision process), which can be viewed as a specialisation of a POMDP... / I think the method would be better presented as reward-modeling for POMDPs... / I think the idea of using LSTMs specifically to handle non-markovian reward is simply a specialisation of this existing idea of using RNNs to encode past observations in POMDPs.*
>
> **A1.** We are sympathetic to your suggestion that our use of LSTMs to handle non-Markovian rewards could be framed as a specialisation of the existing idea to use RNNs to encode past observations in POMDPs. While it is probably true that we could have phrased our entire paper in more general and abstract terms, our primary contribution is precisely that of naming and formalising non-Markovian reward modelling as an interesting and previously-unexplored special case, with great relevance to the problem of AI alignment and human-in-the-loop learning (see Appendix B for a discussion of potential use cases). While our initial set of model architectures provide a starting point which may be familiar to those with experience of POMDP learning, we do employ several techniques inspired by the isomorphism we identify with multiple instance learning (the particular way in which the labelling loss is computed and backpropagated during training, the use of a concatenated skip connection to promote a simple hidden state, the focus on interpretability for understanding and probing the hidden state dynamics). The use of LSTM for POMDPS is referenced in the paper in Section 2, Lines 74 and 75. If there are additional points/references that you think should be included, we welcome further suggestions from you during the discussion period.
>
> **Q2.** *The abstract uses a lot of specialized language RM, MIL,...*
>
> **A2.** Thanks for this feedback and the specific example given; we'll find a clearer way of phrasing the content of the abstract.
>
> **Q3.** *I find it awkward that the reward is a function of both $s_t$ and $h_{t+1}$. I realize that it leads to better performance in practice, but it shouldn’t be necessary and maybe drop it from the math.*
>
> **A3.** We have some sympathy with this view, and debated this notation for a while before settling on the current presentation. We felt that the inclusion of both $s_t$ and $h_{t+1}$ as arguments leads more naturally to the introduction of concatenated skip connections (which, as you say, markedly improves performance), and also helps to reinforce the idea that (unlike in a general POMDP) non-Markovian RM contains a clean separation between state information "out there" in the environment and hidden information "in the head" of the human, both of which have a predictive effect on the reward. Notational issues like this are always tricky, but on balance we still think this is the better option.
>
> **Q4.** *Line 229 mention an “interesting” case when CSC and instance-space LSTM beats the oracles; the reviewer does not think it is interesting but rather suspicious. This could be stemming from a lack of hyperparameter search or uncertainty in the results.*
>
> **A4.** As we allude to in Section 4.3, we suspect that the learnt hidden states may be easier to exploit for policy learning than the raw oracle timer states. This is reinforced by Figure 7, where we show that the learnt hidden states are nonlinear with respect to time and are sparse around an inflection point very close to $t=50$. This likely helps the agent's value network to distinguish this critical timestep in comparison to the linear encoding given in the oracle hidden states. Regarding uncertainty in the results, in Figure 5, we observe that there is no overlap between the lower quartile of the CSC Instance Space LSTM method and the upper quartile of the oracle method, suggesting that this difference is indeed significant. Regarding hyperparameter tuning, although this wasn't a major focus in our work we note that the values given in Appendix E are very much standard for environments of this size, and the same values are used throughout (so the runs using our models were given no inherent advantage through extra tuning).

---

> > ### Comment · Reviewer_2P9E · 2022-08-03
> > **Mostly satisfied with the answers and changes**
> >
> > Thank you to the authors for their thorough response. I'm mostly satisfied with the answers given, with some reservations remaining about the POMDP vs. NMRDP distinction (which are minor and do not warrant a rejection).
> >
> > My remaining comment is about A1 and A3. You talk about the << clean separation between state information "out there" in the environment and hidden information "in the head" of the human, both of which have a predictive effect on the reward. >>, which I agree makes a strong case for the NMRDP formulation, but your examples are non-markovian not because of the human aspect, but because of the partial-observability aspect; the time since the beginning of the episode is not part of the observations yet is necessary for reward computation. This includes the new proposed experiment using lunar lander. These timer-based NMRDP, in my opinion, have no practical applications because it is trivial to add a timer to the observation space and make the task markovian (both in simulation and in real-world deployments). The key and charger examples can be similarly solved trivially by augmenting the observation with easily measurable information. My question is this:
> >
> > Are there interesting examples (where simply adding a timer to the observation does not make the reward markovian) of NMRDPs which are not full-on POMDPs, besides human-assigned rewards?
> >
> > I do not think this is a deal-breaker for this paper, because the paper is focused on reward-modeling where the non-markovian rewards come from imperfect human labeling. But because the experiments do not match that claimed application, this question begs asking.

---

> > > ### Author Response · Authors · 2022-08-04
> > > **Continued Author Response to Reviewer 2P9E**
> > >
> > > Thank you for your comments and facilitating a discussion. Please see our responses below.
> > >
> > > **Q1.** *These timer-based NMRDP, in my opinion, have no practical applications because it is trivial to add a timer to the observation space and make the task markovian (both in simulation and in real-world deployments). The key and charger examples can be similarly solved trivially by augmenting the observation with easily measurable information.*
> > >
> > > **A1.** We agree that the fact that we know the internal structure of our oracles means that it is trivial to augment the observation space with (for example) the timer value, but the point of our experiments was to treat the oracles *as if* they were black boxes and evaluate how well their structure could be recovered from data. Knowing the ground truth was essential for such an evaluation. As for the other implication of your comment - whether our timer, moving, key, and charger oracles have any relevance to our ultimate goal of modelling non-Markovian aspects of human preferences - we would argue that they do. Although these oracles are simple and heavily abstracted, the underlying concepts of gradual accumulation, thresholds, and temporal if-then dependencies (e.g., if you collect the key, then you can access the treasure) could actually serve as valuable building blocks for modelling the affective or preferential dynamics occurring within a human evaluator's mind. Certainly, for any interesting problem, such building blocks would be combined in complicated ways, but in presenting and evaluating the foundations of our approach we feel it was advantageous to strip the concept of a hidden state back to its basics.
> > >
> > > **Q2.** *Are there interesting examples (where simply adding a timer to the observation does not make the reward markovian) of NMRDPs which are not full-on POMDPs, besides human-assigned rewards?*
> > >
> > > **A2.** We consider this to encompass any task with sequential dependency for which we do not know *a priori* how to augment the state space to make it Markovian. This might be because the nature of the sequential dependency is somehow hidden from us, for simply too complicated to encode manually. We think the human-centric framing is most interesting though, which is why we use it in the introductory sections of our paper.

---

### Author Response · Authors · 2022-08-02
**General Rebuttal (1/2)**

We would like to thank all reviewers for their comments. We felt the reviews were fair as well as constructive, and form a solid basis for discussion that will help to improve both this paper and our future work. In this general comment, we summarise reviewers' positive comments about our paper, address the common concerns raised, and enumerate changes that we will make to the paper during the discussion phase. In the specific comments attached to each review, we address issues raised only by individual reviewers. We hope this separation of the common and individual issues aids clarity.

**Summary of positive comments**
All reviewers agree that we have identified a noteworthy gap in the existing reward modelling literature, provided a theoretically sound formalisation, made a novel and valuable connection to multiple instance learning, presented a well-constructed selection of architectures and baselines, and performed a valid preliminary evaluation on bespoke benchmarks tasks. They have praised both the clarity of our writing  and the quality of our figures, and appreciate the inclusion of a qualitative interpretability analysis alongside quantitative performance metrics.

**Common concern: evaluation in more complex environments**
All reviewers raised concerned with regards to the complexity of our experiments. These included comments that the environments used are simple and artificial, do not test our approach on larger-scale real-world tasks, and that we have not applied our methods to common, existing baselines.

To the best of our knowledge, no agreed-upon benchmarks for non-Markovian tasks exist (e.g., all standard OpenAI Gym environments are Markovian), and prior work commonly focuses on grid worlds with very small discrete state spaces [1,2]. Therefore, we believe the continuous environments that we propose in this paper should be viewed as a contribution, rather than a weakness. We specifically designed these environments to be baselines that capture various kinds of non-Markovian structure (binary vs continuous hidden states, time-dependent vs state-dependent hidden state dynamics), meaning they could provide a valuable testbed for future methods. Restricting the state space to be two-dimensional enables interpretable visualisation of the dynamics of algorithms that are targeted at non-Markovian environments.

However, we agree that evaluation in more complex settings is important to demonstrate the wider applicability of our methods. In our first planned revision of this work (see below) we will add this extension to our Limitations and Future Work (Section 5.3). Furthermore, we are making use of the current rebuttal and discussion period to run experiments on an adapted version of the LunarLander environment from OpenAI Gym, with a custom non-Markovian reward function. We hope that this experiment strikes the right balance between satisfying the reviewers that our method does indeed scale to more complex environments (LunarLander has an 8D continuous state space and episodes will run for up to 500 timesteps), and what is practically feasible given the time available. This experiment will be added to a revised version of the paper over the coming days.

**Common concern: evaluation via human experiments**
We agree with the reviewers' comments that running experiments in realistic human-in-the-loop reward modelling scenarios (where labels are generated by people rather than by an oracle) is a major area of future work, which we highlighted in Section 5.3 of the paper. We also note that numerous highly-cited papers in the reward modelling literature use oracle experiments as their sole evaluation method, since it enables scalable quantitative validation [3,4,5].

There are three concrete differences between our oracle preference labelling method and realistic human labelling: 1) preference form, 2) preference sparsity, and 3) preference noise. Preference form captures the different approaches to providing feedback other than direct return labels (e.g., pairwise rankings or good/bad/neural labels), preference sparsity occurs when generating human labels is expensive (whereas oracle labels are cheap), and preference noise arises due to uncertainty in human labels (as opposed to perfect oracle labels).

We decided to focus on noise in this work as it is an established way of making oracle experiments more realistic [6], and is aligned with our discussions of human uncertainties and cognitive biases. Crucially, our experiments in Section 4.4 highlight that our methods degrade gracefully in the presence of noise, which gives us some confidence that they will transfer well to human labels. We are in complete agreement that future work should consider preference sparsity and form, along with evaluations involving actual human data. For better clarity on the subject, we are happy to add a discussion to our Appendix on the differences between our oracle labelling and true human labelling.

---

> ### Author Response · Authors · 2022-08-02
> **General Rebuttal (2/2)**
>
> **Revisions**
> Going forward, we plan to submit a first revision of the paper based on reviewer feedback (written changes only). Following on from that, we will then submit a further revision with an additional experiment (written and experimental changes, including an update to the supplementary material).
>
> **Revision One**
> We will make the following revisions to our paper in light of reviewer feedback (ordered by appearance in the paper):
> 1. Reduce specialised language in abstract.
> 2. State that our results given in Table 1 come from the test set.
> 3. Add the need for more complex environments to Limitations and Future Work.
> 4. Discuss the differences between oracle and human labelling in the Appendix.
>
> **Revision Two**
> We will run an additional experiment involving a more complex task. Our chosen environment is an adapted version of the LunarLander environment from Open AI Gym, in which the lander must first land, and then hover. This two-stage formulation makes the task non-Markovian. We aim for this to demonstrate that our methods scale to more complex environments with higher-dimensional state and action spaces, as well as longer episodes.
>
> **References**
> 1. Littman, Michael L., et al. ``Environment-independent task specifications via GLTL." arXiv preprint arXiv:1704.04341 (2017).
> 2. Gaon, M., and Brafman, R. ``Reinforcement learning with non-markovian rewards". AAAI Conference on Artificial Intelligence 34 (2020).
> 3. Griffith, Shane, et al. ``Policy shaping: Integrating human feedback with reinforcement learning." Advances in Neural Information Processing Systems 26 (2013).
> 4. Hadfield-Menell, Dylan, et al. ``Inverse reward design." Advances in Neural Information Processing Systems 30 (2017).
> 5. Reddy, Siddharth, et al. ``Learning human objectives by evaluating hypothetical behavior." International Conference on Machine Learning. PMLR, 2020.
> 6. Lee, K., et al. ``B-Pref: Benchmarking Preference-Based Reinforcement Learning." Advances in Neural Information Processing Systems 34 (2021).

---

> > ### Comment · Reviewer_2P9E · 2022-08-03
> > **Awaiting the two revisions**
> >
> > I am writing this comment to acknowledge the response to the reviewers, and that I am awaiting the two described revisions. I am satisfied with the proposed changes, and if all goes according to what's described, I will change my score to an "Accept".

---

> > > ### Author Response · Authors · 2022-08-09
> > > **Please see our final revision**
> > >
> > > Thank you again for your comments. Please see our recently submitted final revision, which completes both sets of changes that we laid out in our General Rebuttal.

---

### Author Response · Authors · 2022-08-04
**Revision One**

We have uploaded our first revision of the paper according to our planned changes (please see General Rebuttal Comments below).
The changes are as follows:
1. Reduced the use of specialised language in the abstract.
2. Stated that our results given in Table 1 come from the test set.
3. Added a note on the need for more complex environments to Limitations and Future Work (Section 5.3).
4. Discussed the differences between oracle and human labelling in the Appendix (Appendix F).

For added clarity, we have attached the Appendix to the main body of the paper (previously it was only in the supplementary material).
We are continuing with our ongoing changes for the second revision (additional experiments; please see General Rebuttal Comments below).

---

### Author Response · Authors · 2022-08-09
**Revision Two**

We have uploaded a draft of our second revision of the paper according to our planned changes (please see General Rebuttal Comments below). This revision demonstrates that our approach works on more complex environments (as was requested by all three reviewers). Please note that this is a draft revision - we felt it was best to submit this prior to the final deadline in order to facilitate further discussion before submitting our final version of the paper. The changes are detailed below (note these are in addition to the changes made for Revision One). We also provide information as to what will change between this submission and the final submission. Pending the further minor changes for the final submission, we have achieved everything we set out to in our planned changes (see General Rebuttal Comments below).

**Paper Changes**
1. Added experiments using a more complex task - an adapted version of the Lunar Lander Open AI Gym Task where the agent must land, wait for 50 timesteps, then hover. This is more complex than our previous tasks as it uses a higher dimensional state space (8D rather than 2D), and runs for more timesteps (500 rather than 200).
2. Added reward reconstruction results for the Lunar Lander task in Table 1. We find our MIL models do indeed scale to this more complex task, and exhibit similar trends to our existing results.
3. Added results for RL agent training on the Lunar Lander task to Figure 5. This demonstrates improved RL agent performance compared to the oracle baseline. Please note this figure is still pending further updates as some models are still training.
4. A detailed discussion of our new experiments in Appendix D, including further suggestions of how our approach could be improved.

**Further planned changes for final submission**
1. Update Figure 5 with complete results once the final RL training runs are complete.
2. Make any further changes based on reviewer feedback.
3. Update supplementary material.

---

### Author Response · Authors · 2022-08-09
**Final Revision**

We have now submitted our final revision of the paper. Below we detail the overall changes between this final version and the **original** version. Please note this encapsulates and adds to the changes of our two previous revisions. In this final submission, we have achieved everything we set out in our planned changes (please see the General Rebuttal comments below). The final revised version has five additional pages of appendices compared to the original version. This is due to new experiments, discussions, and clarifications as requested by the reviewers. We also include details below on how we plan to utilise the additional content page for the camera ready version.

**Overall List of Paper Changes**

*New Experiments*
* Added experiments using a more complex task - an adapted version of the Lunar Lander Open AI Gym Task where the agent must land, wait for 50 timesteps, then hover. This is more complex than our previous tasks as it uses a higher dimensional state space (8D rather than 2D), and runs for more timesteps (500 rather than 200).
* Added reward reconstruction results for the Lunar Lander task in Table 1. We find our MIL models do indeed scale to this more complex task, and exhibit similar trends to our existing results.
* Added results for RL agent training on the Lunar Lander task to Figure 5. This demonstrates improved RL agent performance compared to the oracle baseline, again following similar trends to our existing results.

*Additional Appendices*
* Discussed the differences between oracle and human labelling in the Appendix C.
* Added a detailed discussion of our new experiments in Appendix E, including further suggestions of how our approach could be improved for this task.
* Conducted a further analysis of the RL training results for the new Lunar Lander task in Appendix G.
* Included an investigation of the learnt hidden state embeddings for the new Lunar Lander task in Appendix H.

*Minor changes*
* Reduced the use of specialised language in the abstract.
* Stated that our results given in Table 1 come from the test set.
* Added a note on the need for more complex environments to Limitations and Future Work (Section 5.3).

**Planned Changes for Camera Ready Version**
* Replace Figure 4 with Figure A1 (showing all the task environments in the main body of the paper rather than in the Appendix).
* Include hidden state analysis of the new Lunar Lander task in the main body (i.e., move Figure A4 into the main body of the text).
* Expand our interpretability analysis of the Lunar Lander environment with probe trajectory plots (as we did for the other tasks).
* Move certain parts of our discussions from the Appendix into the main body to aid the narrative of the paper
(for example, expanding our Limitations and Future Work section with content currently in the Appendix).

Finally, we would like to thank everyone involved in reviewing this paper for reading and engaging with our work!

---

### Meta-Review · Area_Chair_3pgo · 2022-08-30

**Recommendation:** Accept
**Confidence:** Less certain

**Metareview:**

The reviewers have agreed on many points (at least after some help from the author's explanations and changes in the rebuttal): the problem formulation is interesting (in particular as it relates to evolving human preferences, but also in the practical experimental cases), the writing is clear and the technical solutions are interesting.

While there is also a general consensus that more, larger experiments would be desirable, I note this is much more difficult to achieve in the paper's setup than most "vanilla=Markov" RL, as significant modifications are needed to any standard environment to fit this paradigm. Lunar Lander was well appreciated during the rebuttal, and I believe the paper will now have a strong impact as-is (although if the authors can find the time for another similarly sized env prior to the final version, it will be welcome).

**Award:**

No

---

### Decision · Program_Chairs · 2022-09-14

Accept